# USimUL: Closing the Privacy Gap in Similarity-Based Weakly Supervised Learning

## Abstract

Existing similarity-based weakly supervised learning approaches often rely on precise similarity annotations between data pairs, which may inadvertently expose sensitive label information and raise privacy risks. To mitigate this issue, we propose Uncertain Similarity and Unlabeled Learning (USimUL), a novel framework where each similarity pair is embedded with an uncertainty component to reduce label leakage. In this paper, we propose an unbiased risk estimator that learns from uncertain similarity and unlabeled data. Additionally, we theoretically prove that the estimator achieves statistically optimal parametric convergence rates. Extensive experiments on both benchmark and real-world datasets show that our method achieves superior classification performance compared to conventional similarity-based approaches. Our source code is available at the anonymous link: `https://anonymous.4open.science/r/USimUL-B337`

## 1 Introduction

In supervised classification, the acquisition of precisely labeled data often faces significant challenges in many real-world applications due to privacy regulations and high annotation costs (Bao et al., 2018; Cao et al., 2021; Shi et al., 2024; Wei et al., 2023b; Li et al., 2024). To alleviate this issue, various weakly supervised learning paradigms have emerged as promising alternatives, including but not limited to concealed label learning (Li et al., 2024), semi-supervised learning (Miyato et al., 2018; Lucas et al., 2022; Bai et al., 2024), positive-unlabeled learning (Kiryo et al., 2017; Bekker & Davis, 2020; Zhao et al., 2023; Wang et al., 2024), noisy-label learning (Wang et al., 2019; Han et al., 2020; Wan et al., 2024), partial-label learning (Lv et al., 2020; Zhang et al., 2021; Jia et al., 2024), complementary-label learning (Feng et al., 2020; Xu et al., 2020; Gao & Zhang, 2021; Wei et al., 2023a), and similarity-based classification (Bao et al., 2018; Shi et al., 2024; Li et al., 2025). Among these weakly supervised learning methods, similarity-based classification focus on training a binary classifier by leveraging pairwise similarity labels or similarity-confidence scores instead of explicit pointwise labels. These similarity-based labels indicate whether two instances belong to the same class (similar) or different classes (dissimilar) (Bao et al., 2018; Cao et al., 2021; Wang et al., 2023). Such approaches are particularly useful when collecting fully supervised positive and negative samples is costly or impractical.

However, similarity pairs used in conventional similarity-based learning may inadvertently expose sensitive label information (Cao et al., 2021), which is not naturally appear in other weakly supervised settings. As illustrated in Figure 1 (a), given a similarity pair, if the class label of either instance in a labeled pair is exposed, the label of the other instance can be immediately inferred or estimated, further compromising data privacy. For example, if two individuals (such as police officers) are linked via a similarity association, revealing the label of one may inadvertently disclose sensitive attributes of the other, including identity, affiliation, or income level. This issue becomes particularly critical in high-stakes domains such as healthcare, finance, or national security. Existing similarity-based methods are limited in addressing this risk, as they rely on deterministic pairwise associations that are inherently susceptible to label inference.

To mitigate this issue, we propose Uncertain Similarity and Unlabeled Learning (USimUL), a novel setting that introduces uncertainty into similarity supervision by transforming pairs into triplets. Specifically, as illustrated in Figure 1 (b), we introduce an additional unlabeled instance to the original similarity pair, forming an extended triplet. For example, introducing a civilian into a similarity

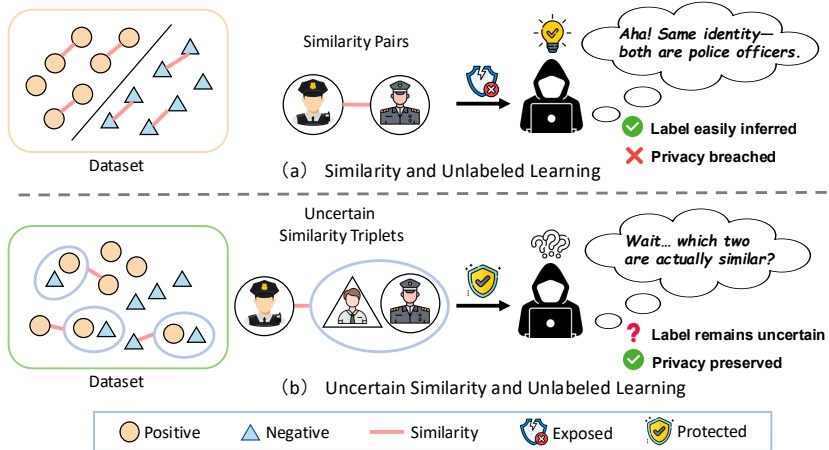

Figure 1: Illustration of label inference risks under different similarity settings. In the traditional similarity pair setup (above), revealing the label of one instance enables deterministic inference of the other's label, compromising privacy. In contrast, USimUL (below) introduces an unlabeled third instance to form an uncertain similarity triplet, preventing label inference and preserving privacy.

relation initially defined between two police officers effectively disrupts the deterministic linkage, thereby ensuring that even if one individual's identity is exposed, the identities of the remaining entities remain indeterminate. Accordingly, USimUL leverages uncertainty as a built-in privacy-preserving mechanism during data annotation, without requiring external encryption or label obfuscation techniques.

In this work, we propose an unbiased risk estimator for learning from uncertain similarity and unlabeled data, and establish a prototype baseline for this novel setting. Theoretically, we derive an upper bound on the evaluation risk and prove that the empirical risk converges to the true classification risk as the number of training samples increases. To validate the effectiveness of our method, we conduct extensive experiments on widely-used benchmark datasets as well as real-world privacy-sensitive datasets and compare its performance against state-of-the-art methods. The primary contributions of this paper are as follows:

- We propose a novel similarity-based weakly supervised learning setting that introduces uncertainty into similarity pairs to prevent privacy leakage.
- We design a simple yet highly effective unbiased framework tailored for this labeling setting. Furthermore, we theoretically analyze and derive the estimation error bound of the proposed method, which demonstrates that the proposed method can converge to the optimal state.
- Extensive experiments on benchmark and real-world datasets validate the superior performance of our method.

## 2 RELATED WORK

**Privacy Labels Learning.** To mitigate privacy concerns during instance-level annotation, recent studies have explored various privacy-aware weak supervision paradigms, including Concealed Label Learning (Li et al., 2024), Label Proportion Learning and Complementary Label Learning (Ishida et al., 2019; Chou et al., 2020; Feng et al., 2020; Xu et al., 2020; Gao & Zhang, 2021). Concealed Label Learning is a novel privacy-preserving setting that aims to protect sensitive labels during the annotation process (Li et al., 2024). Label Proportion Learning (Chai & Tsang, 2022; Patrini et al., 2014; Yu et al., 2013) offers an alternative approach by annotating the proportion of positive instances within a group (or bag), instead of providing explicit labels for individual samples. Complementary Label Learning (Ishida et al., 2019; Xu et al., 2020; Gao & Zhang, 2021; Wei et al., 2023a) is another widely adopted privacy-preserving setting, where each instance is labeled with a class it does not belong to. However, existing privacy-label methods primarily focus on individually labeled samples. Their supervision is attached to individual instances and does not create determin-

istic dependencies across samples, and thus fails to model relational structures such as similarity pairs or triplets.

**Similarity and Unlabeled Learning.** Another line of related work explores the Similarity and Unlabeled Learning (SUL) paradigm (Lu et al., 2019; Cao et al., 2021; Feng et al., 2021; Li et al., 2025). As a foundational contribution, Bao et al. (Bao et al., 2018) demonstrated that empirical risk minimization can be achieved using only similar instance pairs and unlabeled data. Building upon this, Similarity-Confidence Learning (Sconf) (Cao et al., 2021) extended the framework by replacing binary similarity labels with soft confidence scores that reflect pairwise class agreement probabilities. Subsequent advancements introduced learning from confidence difference (ConfDiff) (Wang et al., 2023) or confidence comparison (Pcomp) data (Feng et al., 2021). Recent methods further improve robustness in this context. For example, Robust AUC Maximization (Shi et al., 2024) proposed a framework tailored to Pcomp data, incorporating pairwise surrogate losses that reduce sensitivity to skewed class distributions. Additional extensions such as PCU (Li et al., 2025) aim to enhance stability and learning efficiency under SUL settings. Despite these developments, SUL-based approaches face critical privacy leakage risk. Given a high-confidence similarity pair, if the class label of either instance is exposed, the label of the other can often be inferred. (A comparison with these SUL-based baselines is provided in Appendix I.) This concern motivates us to explore a novel setting that introduces an uncertainty component into the similarity-based pairs to mitigate privacy leakage.

## 3 METHODOLOGY

In this section, we formally define the learning framework for uncertain similarity and unlabeled data, focusing on constructing an unbiased risk estimator. Additionally, we introduce a corrected risk estimator to ensure non-negativity and establish the estimation error bound for our method.

### 3.1 PRELIMINARIES

**Ordinary Classification.** Suppose that $\mathcal{X} \subset \mathbb{R}^d$ is the instance space, and $\mathcal{Y} = \{+1, -1\}$ is the label space. The sample $(x, y) \in \mathcal{X} \times \mathcal{Y}$ are independently sampled from a joint probability distribution with density $P(x, y)$. The objective is to learn a binary classifier $f : \mathcal{X} \to \mathbb{R}$ that minimizes the following classification risk:

$$R(f) = \mathbb{E}_{(x,y) \sim P}[\ell(f(x), y)], \tag{1}$$

where $\mathbb{E}_{(x,y) \sim P}$ denotes the expectation over the joint distribution $P(x, y)$ and $\ell(\cdot, \cdot) : \mathbb{R} \times \mathcal{Y} \to \mathbb{R}^+$ represents a binary loss function. Let $\pi_+ = P(y = 1)$ and $\pi_- = P(y = -1)$ denote the class prior probabilities for the positive and negative classes, respectively. Moreover, let $P_+(x) = P(x \mid y = +1)$ and $P_-(x) = P(x \mid y = -1)$ represent the class-conditional probability densities of positive and negative samples, respectively. Under these definitions, the classification risk in Eq. (1) can be rewritten as

$$R(f) = \mathbb{E}_{P_+(x)} \pi_+ [\ell(f(x), +1)] + \mathbb{E}_{P_-(x)} \pi_- [\ell(f(x), -1)]. \tag{2}$$

**Similarity-based Classification.** Recently, many studies have tried to solve the similarity-based and unlabeled learning (SUL) problem (Bao et al., 2018; Feng et al., 2021; Cao et al., 2021; Wang et al., 2023). Let $(x, x')$ denotes a similar data pair, where both instances belong to the same class. The goal of SUL is to learn a classifier using only similarity and unlabeled data, eliminating the need for fully labeled datasets. Unfortunately, these studies fail to account for the significant privacy risks involved: if the class label of either $x$ or $x'$ is exposed, the label of the paired instance is also revealed. This risk becomes critical when data contain sensitive attributes (e.g., racial identity and religious orientations), potentially leading to privacy leakage.

### 3.2 UNCERTAIN SIMILARITY AND UNLABELED LEARNING

To mitigate the risk of privacy leakage, we propose a novel weakly supervised learning framework, **U**ncertain **Sim**ilarity and **U**nlabeled **L**earning (**USimUL**). Specifically, we introduce an additional unlabeled instance $x''$ into the similarity pair $(x, x')$, forming an extended triplet $(x, \{x', x''\})$ that disrupts direct pairwise associations. As illustrated in Figure 1, the disclosure of a single instance's

label does not compromise the privacy of the remaining instances. To derive an unbiased risk estimator, we first establish a rigorous formulation of the generation of uncertain similarity data and introduce the following formal definition.

**Definition 1 (Uncertain Similarity Triplet).** *A triplet $(x, \{x', x''\})$ is sampled such that two out of the three instances share the same class label, but it is unknown which two. The formation of uncertain similarity triplets follows:*

$$P_{US}\left(x, \{x', x''\}\right) = P(x, x', x'' \mid (y = y' = 1) \text{ or } (y = y' = -1) \text{ or } (y = y'' = 1) \text{ or } (y = y'' = -1)). \tag{3}$$

The uncertain similarity triplet $(x, x', x'')$ introduces ambiguity into traditional pairwise similarity by relaxing the requirement that both associated instances share the same label. Instead, it ensures that at least two out of the three instances belong to the same class, but it is unknown which pair. More concretely, the triplet is sampled from $P_{US}(x, \{x', x''\})$ such that one of the following conditions holds: $x$ and $x'$ belong to the same class (either positive or negative), or $x$ and $x''$ belong to the same class.

**Superiority of Uncertain Similarity Data.** This construction prevents direct inference of individual labels and thus weakens deterministic linkages inherent in traditional similarity pairs. From a learning perspective, it allows the model to still benefit from similarity information while introducing uncertainty that mitigates the risk of label leakage.

**Unbiased Risk Estimator with USimU Data.** In this section, we derive an unbiased estimator of the classification risk in Eq. (1) using uncertain similarity triplets and unlabeled data (USimU data), and we establish its risk minimization framework. Firstly, we formally denote the set of uncertain similarity triplets as $\mathcal{D}_{US}$ and the set of unlabeled instance as $\mathcal{D}_U$, given by:

$$\mathcal{D}_{US} \triangleq \left\{ \left(x_i, \{x_i', x_i''\}\right) \right\}_{i=1}^{N_{US}} \overset{i.i.d.}{\sim} P_{US}(x, \{x', x''\}), \qquad \mathcal{D}_U \triangleq \{x_i\}_{i=1}^{N_U} \overset{i.i.d.}{\sim} P_U(x), \tag{4}$$

where $N_{US}$ and $N_U$ denote the number of uncertain similarity triplets in $\mathcal{D}_{US}$ and the unlabeled instances in $\mathcal{D}_U$. We also define $\widetilde{\mathcal{D}}_{US} \triangleq \{x_i\}_{i=1}^{3N_{US}} \overset{i.i.d.}{\sim} \widetilde{P}_{US}(x)$ as the pointwise uncertain similarity dataset, obtained by disregarding the triplet structure in $\mathcal{D}_{US}$. Our goal is to learn a classifier only from USimU data.

In Eq. (3), the conditional distribution $P(x, x', x'' \mid (y = y' = 1) \text{ or } (y = y' = -1) \text{ or } (y = y'' = 1) \text{ or } (y = y'' = -1))$ is not directly available for training. To address this, we express it as:

$$P(x, x', x'' \mid Y) = \frac{P(x, x', x'', Y)}{P(Y)}, \tag{5}$$

where $Y = \{(y = y' = 1) \text{ or } (y = y' = -1) \text{ or } (y = y'' = 1) \text{ or } (y = y'' = -1)\}$. Fortunately, both $P(x, x', x'', Y)$ and $P(Y)$ can represented by introducing the class priors $P(y = 1)$ and $P(y = -1)$. For tractability, we assume that samples within each triplet are independently drawn. While this assumption may not hold strictly in real-world settings, we argue that it provides a useful approximation for theoretical analysis, consistent with prior work in weakly supervised learning (Bao et al., 2018; Feng et al., 2021; Cao et al., 2021).

**Lemma 2.** *Given the class priors $\pi_+ = P(y = 1)$ and $\pi_- = P(y = -1)$, and assuming that $x$, $x'$, and $x''$ are mutually independent, $P(x, x', x'', Y)$ and $P(Y)$ can be expressed as:*

$$P(x, x', x'', Y) = 2\left[\pi_+^2 P_+^2(x) + \pi_-^2 P_-^2(x)\right]P(x), \qquad P(Y) = 1 - \pi_+\pi_-, \tag{6}$$

*where $P_+(x) = P(x \mid y = +1)$ and $P_-(x) = P(x \mid y = -1)$ denote the class-conditional probability densities of positive and negative samples, respectively, and $P(x)$ denotes the marginal density over all samples.*

The proof is provided in the Appendix A. Lemma 2 states that both $P(x, x', x'', Y)$ and $P(Y)$ can be expressed in terms of the class priors $P(y = 1)$ and $P(y = -1)$. This lemma provides the probabilistic foundation for modeling uncertain similarity triplets by expressing the joint probability of triplet instances in terms of class priors and conditional probabilities. Building on Lemma 2 and the definition of $\widetilde{\mathcal{D}}_{US} \triangleq \{x_i\}_{i=1}^{3N_{US}} \overset{i.i.d.}{\sim} \widetilde{P}_{US}(x)$, we establish the following lemma.

**Lemma 3.** *The dataset $\widetilde{\mathcal{D}}_{US} \triangleq \{\widetilde{x}_i\}_{i=1}^{3N_{US}}$ consists of independently drawn samples following:*

$$\widetilde{P}_{US}(x) = \frac{2\left[\pi_+^2 P_+(x) + \pi_-^2 P_-(x)\right]}{1 - \pi_+\pi_-}. \tag{7}$$

The proof is provided in the Appendix B. Lemma 3 establishes that each instance in a triplet $(x, x', x'')$ is marginally distributed according to $\widetilde{P}_{US}(x)$ (Eq. (7)), enabling pointwise risk estimation despite the triplet structure. This perspective is crucial for deriving the unbiased risk estimator. Next, we reformulate the classification risk in Eq. (2) with only USimU data. Assume $\pi_+ \neq \frac{1}{2}$, given the class priors $\pi_+ = P(y = 1)$ and $\pi_- = P(y = -1)$, we define the parameters $\theta_{US}^+, \theta_{US}^-, \theta_U^+$, and $\theta_U^-$ as follows:

$$\theta_{US}^+ = \frac{1 - \pi_+ \pi_-}{2(\pi_+ - \pi_-)}, \qquad \theta_{US}^- = \frac{1 - \pi_+ \pi_-}{2(\pi_- - \pi_+)}, \qquad \theta_U^+ = \frac{-2\pi_-}{2(\pi_+ - \pi_-)}, \qquad \theta_U^- = \frac{-2\pi_+}{2(\pi_- - \pi_+)}. \quad (8)$$

Subsequently, by utilizing Eq. (8) to reformulate the classification risk, we derive the following theorem.

**Theorem 4.** *The classification risk can be equivalently expressed as*

$$R_{USU}(f) = \mathbb{E}_{x \sim \widetilde{P}_{US}(x)} \left\{ \bar{\ell}_+[f(x)] \right\} + \mathbb{E}_{x \sim P_U(x)} \left\{ \bar{\ell}_-[f(x)] \right\}, \quad (9)$$

*where $\bar{\ell}_+(z) = \theta_{US}^+ \ell(z, +1) + \theta_{US}^- \ell(z, -1)$ and $\bar{\ell}_-(z) = \theta_U^+ \ell(z, +1) + \theta_U^- \ell(z, -1)$.*

The proof is provided in the Appendix C. As we can see from Theorem 4, $R_{USU}(f)$ can be assessed in the training stage only using USimU data. [1]

**Empirical Risk.** Since the training dataset $\widetilde{\mathcal{D}}_{US}$ is sampled independently from the $\widetilde{P}_{US}(x)$, the empirical risk estimator can be naively approximated as:

$$\widehat{R}_{USU}(f) = \frac{1}{3N_{US}} \sum_{i=1}^{3N_{US}} \left\{ \bar{\ell}_+[f(x_i)] \right\} + \frac{1}{N_U} \sum_{j=1}^{N_U} \left\{ \bar{\ell}_-[f(x_j)] \right\}, \quad (10)$$

where $N_{US}$ and $N_U$ denote the number of uncertain similarity triplets in $\mathcal{D}_{US}$ and the unlabeled instances in $\mathcal{D}_U$. The definitions of $\bar{\ell}_+$ and $\bar{\ell}_-$ are provided above. To help non-expert readers better understand the procedure, we present a step-by-step algorithm in Appendix H.

**Corrected Risk Estimator.** Since the classification risk is defined as the expectation of a non-negative loss function $\ell(f(x), y)$, both the risk and its empirical counterpart are lower-bounded by zero, i.e., $R_{USU}(f) \geq 0$ and $\widehat{R}_{USU}(f) \geq 0$. However, similar to issue of the empirical approximator going negative in binary classification from similarity-based methods (Bao et al., 2018; Cao et al., 2021), the empirical risk estimator in Eq. (10) may become negative due to the presence of negative coefficients in the loss formulation.

To address this, enforcing non-negativity of the classification risk has proven effective in weakly supervised learning settings, as demonstrated in prior works (Cao et al., 2021; Feng et al., 2021; Wang et al., 2023). Motivated by this, we propose the following corrected risk estimator specifically tailored for learning from USimU data by applying a correction function to ensure non-negativity.

$$\widehat{R}_{USU}^g(f) = g\left[ \frac{1}{3N_{US}} \sum_{i=1}^{3N_{US}} \left\{ \bar{\ell}_+[f(x_i)] \right\} + \frac{1}{N_U} \sum_{j=1}^{N_U} \left\{ \bar{\ell}_-[f(x_j)] \right\} \right], \quad (11)$$

where $g[z]$ denotes the correction function, such as the max-operator function $g[z] = max\{0, z\}$.

Although using a max-operator in the corrected empirical risk ensures non-negativity within each mini-batch, it introduces a limitation: the risk associated with each label cannot approach zero. This approach effectively ignores the optimization of negative risk values, thereby failing to sufficiently reduce overfitting. To address this limitation, we propose an alternative correction function defined as $g[z] = |z|$, where $|z|$ denotes the absolute value of $z$, i.e., $|z| = max\{0, z\} - min\{0, z\}$. This correction function allows the risk associated with each label to converge toward zero during training, thereby providing a more effective mechanism for mitigating overfitting in uncertain similarity and unlabeled learning.

---

[1]Note that Theorem 4 can be further generalized to handle nonlinear $f$ or arbitrary loss functions $\ell$, as also discussed in prior work (Lu et al., 2019).

### 3.3 CLASS-PRIOR ESTIMATION FROM USIMU DATA

In this section, we propose a class-prior estimation algorithm only from uncertain similarity triplets and unlabeled (USimU) data. First, let us begin with connecting the marginal distribution $P(x, x', x'')$ and $P_{US}(x, x', x'')$ when three examples $x, x', x''$ are drawn independently:

$$P(x, x', x'') = \pi_{US} P_{US}(x, x', x'') + \pi_D P_D(x, x', x''), \tag{12}$$

where $\pi_D = \pi_- \pi_+^2 + \pi_+ \pi_-^2$, and

$$P_D(x, x', x'') = P(x, x'x''|(y = -1 \, and \, y' = y'' = 1) \, or \, (y = 1 \, and \, y' = y'' = -1))$$
$$= \frac{2[\pi_- \pi_+^2 P_-(x) P_+(x') P_+(x'') + \pi_+ \pi_-^2 P_+(x) P_-(x') P_-(x'')]}{1 - \pi_+ \pi_-}. \tag{13}$$

Marginalizing out $x', x''$ in Eq. (12) as Lemma 3, we can obtain $P(x) = \pi_{US} \widetilde{P}_{US}(x) + \pi_D \widetilde{P}_D(x)$, where $\widetilde{P}_{US}$ is define in Eq. (7), and

$$\widetilde{P}_D(x) = \frac{2[\pi_- \pi_+^2 P_-(x) + \pi_+ \pi_-^2 P_+(x)]}{1 - \pi_+ \pi_-}. \tag{14}$$

Since we have samples $\mathcal{D}_U$ and $\widetilde{\mathcal{D}}_{US}$ drawn from $P_U$ and $\widetilde{P}_{US}$ respectively, we can estimate $\pi_{US}$ by mixture proportion estimation methods [2] (Scott, 2015; Sakai et al., 2017; Bao et al., 2018). Then, following $\pi_{US} = 1 - (\pi_- \pi_+^2 + \pi_+ \pi_-^2) = 1 - [(1 - \pi_+)\pi_+^2 + \pi_+(1 - \pi_+)^2] = 1 - \pi_+ + \pi_+^2$, and assuming $\pi_+ > \pi_-$, we obtain $\pi_+ = \frac{\sqrt{4\pi_{US} - 3} + 1}{2}$. We summarize a wrapper of mixture proportion estimation in the Algorithm 2 in appendix.

### 3.4 ESTIMATION ERROR BOUND

Here, the estimation error bound of the proposed unbiased risk estimator is derived to theoretically justify the effectiveness of our method. Let $\mathbf{f} = [f_+, f_-]$ denote the classification vector function in the hypothesis set $\mathcal{F}$. Using $C_\phi$ to denote the upper bound of the $\bar{\ell}_+(z)$ and $\bar{\ell}_-(z)$. Let $L_\phi$ be the Lipschitz constant of $\phi$, we can introduce the following lemma.

**Lemma 5.** *For any $\delta > 0$, with the probability at least $1 - \delta$,*

$$\sup_{\mathbf{f} \in \mathcal{F}} \left| R_{US}(\mathbf{f}) - \widehat{R}_{US}(\mathbf{f}) \right| \leqslant 2L_\phi \Re_{N_{US}}(\mathcal{F}) + C_\phi \sqrt{\frac{2\ln(4/\delta)}{3N_{US}}},$$

$$\sup_{\mathbf{f} \in \mathcal{F}} \left| R_U(\mathbf{f}) - \widehat{R}_U(\mathbf{f}) \right| \leqslant 2L_\phi \Re_{N_U}(\mathcal{F}) + C_\phi \sqrt{\frac{2\ln(4/\delta)}{N_U}},$$

*where $R_{US}(\mathbf{f}) = \mathbb{E}_{x \sim \widetilde{P}_{US}(x)} \bar{\ell}_+[f(x)]$, $R_U(\mathbf{f}) = \mathbb{E}_{x \sim P_U(x)} \bar{\ell}_-[f(x)]$, and $\widehat{R}_{US}(\mathbf{f})$ and $\widehat{R}_U(\mathbf{f})$ denote the empirical risk estimator to $R_{US}(\mathbf{f})$ and $R_U(\mathbf{f})$, respectively. $\Re_{N_{US}}(\mathcal{F})$, and $\Re_{N_U}(\mathcal{F})$ are the Rademacher complexities(Mohri et al., 2018) of $\mathcal{F}$ for the sampling of size $3N_{US}$ from $\widetilde{P}_{US}(x)$ and the sampling of size $N_U$ from $P_U(x)$.*

The proof is provided in the Appendix D. Lemma 5 provides bounds on the difference between the true risk (expected loss) of the classification function $\mathbf{f}$ under two distributions $\widetilde{P}_{US}(x)$ and $P_U(x)$ and their respective empirical risk estimates based on finite samples. This lemma essentially describes how close the empirical risk is to the true risk, with high probability, for any function $\mathbf{f} \in \mathcal{F}$. Based on the Lemma 5, we can obtain the estimation error bound as follows.

**Theorem 6.** *For any $\delta > 0$, with the probability at least $1 - \delta$,*

$$R_{USU}(\hat{\mathbf{f}}) - \min_{\mathbf{f} \in \mathcal{F}} R_{USU}(\mathbf{f}) \leqslant 4L_\phi \Re_{N_{US}}(\mathcal{F}) + 4L_\phi \Re_{N_U}(\mathcal{F}) + 2C_\phi \sqrt{\frac{2\ln(4/\delta)}{3N_{US}}} + 2C_\phi \sqrt{\frac{2\ln(4/\delta)}{N_U}}, \tag{15}$$

---

[2]Given distribution $F$ which is a convex combination of distributions $G$ and $H$ such that $F = (1 - k)G + kH$, the mixture proportion estimation problem is to estimate $k \in [0, 1]$ only with samples from $F$ and $H$. In our case, $F$, $H$, and $k$ correspond to $P$, $\widetilde{P}_{US}$, and $\pi_{US}$, respectively (Scott, 2015; Bao et al., 2018).

Table 1: Classification accuracy of each algorithm on benchmark datasets. We report the mean and standard deviation of results over 5 trials. The best method is highlighted in **bold** and the second-best method is underlined (under 5% t-test).

| Class Prior | Setting | Method | MNIST | Fashion | Kuzushiji | CIFAR-10 | SVHN |
|---|---|---|---|---|---|---|---|
| $\pi_+ = 0.4$ | Baselines | Sconf-ABS | $80.82 \pm 0.57$ | $78.69 \pm 0.53$ | $70.62 \pm 0.77$ | $63.68 \pm 2.30$ | $63.55 \pm 2.17$ |
| | | Sconf-NN | $83.34 \pm 0.55$ | $78.95 \pm 0.26$ | $71.73 \pm 0.84$ | $64.44 \pm 0.11$ | $58.38 \pm 0.27$ |
| | Conf Comparison | Pcomp-ReLU | $87.72 \pm 0.05$ | $87.12 \pm 0.03$ | $84.22 \pm 0.09$ | $72.36 \pm 0.50$ | $71.16 \pm 0.77$ |
| | | Pcomp-ABS | $87.21 \pm 0.04$ | $86.63 \pm 0.53$ | $83.75 \pm 0.38$ | $71.23 \pm 0.66$ | $68.82 \pm 2.37$ |
| | | Pcomp-Teacher | $85.99 \pm 0.28$ | $85.55 \pm 0.20$ | $74.44 \pm 0.81$ | $73.33 \pm 0.08$ | $71.74 \pm 0.06$ |
| | | PC-AUC | $88.52 \pm 0.15$ | $87.80 \pm 0.08$ | $84.53 \pm 0.31$ | $75.07 \pm 0.57$ | $81.33 \pm 0.38$ |
| | | PCU | $83.09 \pm 2.81$ | $86.77 \pm 1.98$ | $81.45 \pm 1.63$ | $80.76 \pm 1.22$ | $79.36 \pm 2.54$ |
| | Conf Difference | ConfDiff-Unbiased | $93.63 \pm 0.12$ | $93.01 \pm 0.19$ | $84.20 \pm 1.06$ | $76.96 \pm 1.69$ | $68.64 \pm 0.90$ |
| | | ConfDiff-ReLU | $93.68 \pm 0.21$ | $92.35 \pm 0.12$ | $84.07 \pm 0.93$ | $82.16 \pm 0.28$ | $84.45 \pm 1.09$ |
| | | ConfDiff-ABS | $94.11 \pm 0.05$ | $92.69 \pm 0.51$ | $85.13 \pm 0.14$ | $82.13 \pm 0.25$ | $82.06 \pm 0.28$ |
| | | **USimUL (Our)** | $\mathbf{95.36 \pm 0.23}$ | $\mathbf{95.51 \pm 0.04}$ | $\mathbf{87.10 \pm 0.30}$ | $\mathbf{84.62 \pm 0.31}$ | $\mathbf{87.18 \pm 0.95}$ |
| $\pi_+ = 0.6$ | Baselines | Sconf-ABS | $83.88 \pm 2.49$ | $79.21 \pm 2.34$ | $69.42 \pm 1.18$ | $64.55 \pm 0.48$ | $60.04 \pm 0.05$ |
| | | Sconf-NN | $82.79 \pm 1.10$ | $80.01 \pm 0.81$ | $70.89 \pm 0.27$ | $62.86 \pm 1.58$ | $61.79 \pm 1.76$ |
| | Conf Comparison | Pcomp-ReLU | $87.44 \pm 0.30$ | $87.12 \pm 0.02$ | $84.14 \pm 0.02$ | $73.94 \pm 0.49$ | $71.80 \pm 0.44$ |
| | | Pcomp-ABS | $84.02 \pm 0.11$ | $87.66 \pm 0.91$ | $80.72 \pm 0.46$ | $72.66 \pm 0.08$ | $71.72 \pm 0.19$ |
| | | Pcomp-Teacher | $85.00 \pm 1.42$ | $82.73 \pm 0.17$ | $75.93 \pm 0.37$ | $75.06 \pm 0.13$ | $72.07 \pm 1.34$ |
| | | PC-AUC | $88.09 \pm 0.15$ | $90.89 \pm 0.15$ | $83.62 \pm 0.14$ | $78.47 \pm 0.05$ | $79.58 \pm 1.02$ |
| | | PCU | $84.08 \pm 1.48$ | $86.00 \pm 6.41$ | $79.99 \pm 2.12$ | $74.31 \pm 5.51$ | $76.66 \pm 3.49$ |
| | Conf Difference | ConfDiff-Unbiased | $93.94 \pm 0.22$ | $91.83 \pm 0.21$ | $86.61 \pm 0.17$ | $78.06 \pm 0.61$ | $68.21 \pm 0.34$ |
| | | ConfDiff-ReLU | $93.58 \pm 0.19$ | $92.88 \pm 0.21$ | $86.65 \pm 0.21$ | $81.38 \pm 0.40$ | $83.23 \pm 0.38$ |
| | | ConfDiff-ABS | $93.97 \pm 0.18$ | $92.61 \pm 0.26$ | $86.60 \pm 0.16$ | $82.78 \pm 1.21$ | $83.89 \pm 2.02$ |
| | | **USimUL (Our)** | $\mathbf{95.05 \pm 0.20}$ | $\mathbf{95.78 \pm 0.06}$ | $\mathbf{88.62 \pm 0.17}$ | $\mathbf{85.22 \pm 0.06}$ | $\mathbf{87.92 \pm 0.12}$ |
| $\pi_+ = 0.2$ | Conf Comparison | Pcomp-ReLU | $90.10 \pm 0.01$ | $92.92 \pm 0.14$ | $82.57 \pm 0.03$ | $80.84 \pm 0.03$ | $80.44 \pm 0.06$ |
| | | Pcomp-ABS | $90.12 \pm 0.13$ | $89.93 \pm 0.03$ | $82.46 \pm 0.02$ | $80.77 \pm 0.73$ | $80.11 \pm 0.17$ |
| | | Pcomp-Teacher | $89.18 \pm 0.01$ | $91.76 \pm 0.02$ | $80.53 \pm 0.04$ | $76.48 \pm 2.13$ | $63.78 \pm 2.45$ |
| | | PC-AUC | $91.95 \pm 0.02$ | $93.28 \pm 0.02$ | $83.60 \pm 0.25$ | $75.69 \pm 1.33$ | $80.01 \pm 0.00$ |
| | | PCU | $84.08 \pm 4.00$ | $90.43 \pm 2.79$ | $81.37 \pm 0.44$ | $79.72 \pm 1.53$ | $78.34 \pm 1.79$ |
| | Conf Difference | ConfDiff-Unbiased | $90.89 \pm 0.12$ | $92.93 \pm 0.01$ | $80.01 \pm 0.12$ | $80.28 \pm 0.48$ | $80.17 \pm 0.02$ |
| | | ConfDiff-ReLU | $80.09 \pm 0.01$ | $80.89 \pm 0.05$ | $80.13 \pm 0.04$ | $81.69 \pm 1.36$ | $80.85 \pm 0.54$ |
| | | ConfDiff-ABS | $80.00 \pm 0.00$ | $80.05 \pm 0.02$ | $80.01 \pm 0.03$ | $81.51 \pm 1.02$ | $80.02 \pm 0.03$ |
| | | **USimUL (Our)** | $\mathbf{94.08 \pm 0.08}$ | $\mathbf{94.50 \pm 0.14}$ | $\mathbf{85.02 \pm 0.38}$ | $\mathbf{83.13 \pm 0.03}$ | $\mathbf{87.65 \pm 0.34}$ |

*where $\hat{\mathbf{f}}$ is trained by minimizing the classification risk $R_{USU}$. The proof is provided in the Appendix E. Lemma 5 and Theorem 6 demonstrate that as the number of USimU data increases, the estimation error of the learned classifiers decreases. When deep network hypothesis set $\mathcal{F}$ is fixed and satisfies the Rademacher complexity bound $\mathfrak{R}_N(\mathcal{F}) \leqslant C_{\mathcal{F}}/\sqrt{N}$, it follows that $\mathfrak{R}_{N_{US}}(\mathcal{F}) = \mathcal{O}(1/\sqrt{N_{US}})$, and $\mathfrak{R}_{N_U}(\mathcal{F}) = \mathcal{O}(1/\sqrt{N_U})$. Consequently, we have:*

$$N_{US}, N_U \to \infty \implies R_{USU}(\hat{\mathbf{f}}) - \min_{\mathbf{f} \in \mathcal{F}} R_{USU}(\mathbf{f}) \to 0$$

Lemma 5 and Theorem 6 theoretically justify the effectiveness of our method for learning from uncertain similarity and unlabeled data, confirming that the proposed method converges to the optimal solution as data size increases.

## 4 EXPERIMENTS

This section provides the primary experimental results and ablation analyses. For further supplementary ablation studies and visualizations, please refer to Appendix F.1–F.5.

### 4.1 EXPERIMENTAL SETUP

**Datasets.** We conduct experiments on five widely used benchmark datasets: MNIST (LeCun et al., 1998), Fashion (Xiao et al., 2017), Kuzushiji (Clanuwat et al., 2018), CIFAR-10 (Torralba et al., 2008), and SVHN (Netzer et al., 2011). Additionally, we evaluate our approach on four real-world weakly supervised learning (WSL) datasets, including Pendigits (Blake, 1998), Lost (Cour et al., 2011), BirdSong (Briggs et al., 2012), MSRCv2 (Liu & Dietterich, 2012). Furthermore, we evaluate our approach on three real-world privacy-sensitive datasets, namely DDSM [3] (Digital Database for

---

[3] DDSM: http://www.eng.usf.edu/cvprg/Mammography/Database.html

Table 2: Classification accuracy of each algorithm on real-world WSL datasets. The best method is highlighted in **bold** and the second-best method is underlined (under 5% t-test, $\pi_+ = 0.4$).

| Dataset | Baselines | | Pcomp | | | ConfDiff | | | PC-AUC | USimUL |
|---|---|---|---|---|---|---|---|---|---|---|
| | Sconf-ABS | Sconf-NN | ReLU | ABS | Teacher | Unbiased | ReLU | ABS | | |
| Pendigits | 77.58±0.10 | 79.22±0.61 | 88.76±0.88 | 88.06±0.34 | 89.60±0.65 | 92.70±0.61 | 93.28±0.31 | 95.24±0.11 | 88.30±0.10 | **97.00±0.41** |
| Lost | 61.93±0.57 | 62.36±0.56 | 73.45±0.42 | 72.89±0.98 | 72.97±1.06 | 64.99±0.92 | 65.17±1.12 | 63.84±0.20 | 76.85±1.56 | **81.46±0.56** |
| MSRCv2 | 63.61±3.22 | 68.18±1.30 | 73.70±1.62 | 69.81±2.27 | 75.65±0.97 | 72.95±0.42 | 73.46±0.55 | 72.08±1.30 | 75.00±0.97 | **77.28±2.60** |
| BirdSong | 66.41±0.78 | 66.79±0.39 | 73.09±1.87 | 75.35±1.32 | 77.06±0.70 | 78.23±1.24 | 78.69±0.78 | 79.66±0.42 | 76.75±1.17 | **81.57±1.63** |

Table 3: Classification accuracy of each algorithm on real-world privacy-sensitive datasets. The best method is highlighted in **bold** and the second-best method is underlined (under 5% t-test, $\pi_+ = 0.4$).

| Dataset | Baselines | | Pcomp | | | ConfDiff | | | PC-AUC | USimUL |
|---|---|---|---|---|---|---|---|---|---|---|
| | Sconf-ABS | Sconf-NN | ReLU | ABS | Teacher | Unbiased | ReLU | ABS | | |
| DDSM | 69.18±0.68 | 66.78±0.36 | 74.91±3.39 | 78.99±0.55 | 71.69±0.85 | 75.34±1.48 | 75.82±1.85 | 75.79±1.16 | 70.89±0.22 | **81.85±0.34** |
| PDMD | 78.00±2.00 | 70.13±1.27 | 82.98±4.55 | 84.41±1.72 | 84.47±2.04 | 86.92±0.32 | 85.63±1.94 | 87.04±1.75 | 76.62±3.27 | **90.00±2.00** |
| PDSD | 75.58±1.03 | 66.28±1.79 | 85.52±1.31 | 82.68±2.85 | 83.72±0.83 | 84.82±2.56 | 82.79±1.28 | 84.79±1.37 | 75.97±3.95 | **91.86±2.16** |

Screening Mammography), PDMD (Privacy Data of Monkeypox Disease), and PDSD (Privacy Data of Skin Disease).

The DDSM dataset consists of a substantial collection of medical images, which contain sensitive information about individual's health status and disease progression. Without proper privacy protection measures, utilizing this dataset for research or analysis could lead to privacy leakage, potentially violating data protection regulations like the GDPR (Kuner et al., 2021). For this reason, we chose the DDSM dataset to evaluate our proposed method. Additionally, we have collected two real-world datasets (PDMD and PDSD) specifically focused on privacy-sensitive disorders, each containing images of both healthy and diseased individuals. For the DDSM, PDMD, and PDSD datasets, each image is resized to $64 \times 64 \times 3$.

Following prior work (Lu et al., 2020; Cao et al., 2021), we manually transform the multi-class datasets into binary classification datasets to maintain consistency across experiments. We follow the original SUL setting to obtain the similarity-based data and then randomly introduce an unlabeled sample to construct an uncertain similarity triplet. Subsequently, we randomly sample a subset of instances from the remaining training data and completely remove their labels to form the unlabeled dataset $D_U$. These unlabeled samples follow the marginal distribution $P(x)$. Further details of the datasets used are provided in the Appendix G.

**Compared Approaches.** To comprehensively evaluate the effectiveness of the proposed method, we compare it against three categories of approaches:

- **Baselines.** The classic similarity-confidence learning baselines, including Sconf-ABS (Cao et al., 2021) and Sconf-NN (Cao et al., 2021).
- **Conf Comparison.** The latest confidence comparison methods, such as Pcomp-ReLU (Feng et al., 2021), Pcomp-ABS (Feng et al., 2021), Pcomp-Teacher (Feng et al., 2021), PC-AUC (Shi et al., 2024), and PCU (Li et al., 2025).
- **Conf Difference.** The state-of-the-art confidence difference methods, including ConfDiff-Unbiased (Wang et al., 2023), ConfDiff-ReLU (Wang et al., 2023), ConfDiff-ABS (Wang et al., 2023).

**Implementation Details.** For Sconf-ABS, Sconf-NN, ConfDiff-Unbiased, ConfDiff-ReLU, ConfDiff-ABS, and PC-AUC, we assign confidence scores or confidence difference scores to each sample in the similarity triplets following the methodology outlined in their respective papers. Note that these confidence scores are not present in our method, which means the aforementioned compared methods use a higher level of supervision information compared to our method. To ensure a fair comparison, we employ the same model across all the compared approaches. All experiments are conducted using PyTorch and executed on a NVIDIA GeForce RTX 4090 GPU. We optimize all compared methods using the same Adam optimizer, with learning rate and weight-decay candidates selected from $\{1, 1e^{-1}, 1e^{-2}, 1e^{-3}, 1e^{-4}, 1e^{-5}, 1e^{-6}\}$. The mini-batch size is set to 256 and the epoch size is set to 100. The hyperparameters for all compared approaches are tuned to maximize test set accuracy.

**Loss Function and Model.** In our experiments, we use the square loss $\phi(z) = (1-z)^2$ as the loss function $\ell$ to train the classifier. Further details of the model used are provided in the Appendix G.

## 4.2 Main Results and Analysis

**Benchmark Datasets.** We evaluate our method on five widely used benchmark datasets: MNIST, Kuzushiji, Fashion, CIFAR-10, and SVHN. As shown in Table 1, the proposed method consistently outperforms existing methods across all benchmark

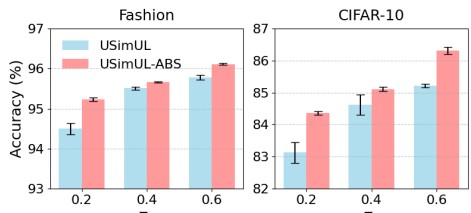

Figure 2: Comparison of the accuracy of USimUL and USimUL-ABS under different class priors.

datasets. Key findings include: i) Compared to classic similarity-confidence learning methods (Baselines), our method demonstrates significant advantages across all experiments. ii) Compared to the state-of-the-art similarity-confidence comparison methods, our method exhibits a noticeable performance improvement. iii) Even when utilizing weaker supervision, our method remains competitive against the most recent Conf-Diff-based methods, achieving state-of-the-art results.

**Real-world WSL datasets.** To assess practical applicability, we further validate our method on real-world weakly supervised learning (WSL) datasets. As shown in Table 2, our method achieves the highest accuracy with minimal variance, consistently outperforming all compared methods on real-world WSL datasets. Notably, the proposed method outperforms the second-best method by $2.67\%$ (Pendigits), $4.57\%$ (Lost), $3.78\%$ (MSRCv2), $3.58\%$ (BirdSong). These results further validate the superior generalization of the proposed method in real-world WSL scenarios.

**Real-world Privacy-Sensitive Datasets.** To further validate the effectiveness of our method, we conduct additional experiments on three real-world privacy-sensitive datasets. Table 3 presents the mean and variance of the prediction accuracy across all comparison methods on these datasets.

The experimental results highlight the significant advantages of the proposed method in most scenarios. Specifically: i) Under the setting of $\pi_+ = 0.4$, our method outperforms all compared methods on the DDSM, PDMD, and PDSD datasets, achieving up to $4.86\%$ improvement over the second-best method. ii) The standard deviation of USimUL is generally lower than the compared methods, indicating our method's stronger stability across different data distributions. This reduced variance is particularly crucial in real-world applications, as it can reduce the risk of performance fluctuations caused by data bias. In summary, our method consistently demonstrates superior performance and stability on real-world privacy-sensitive datasets.

## 4.3 Results of Corrected Risk Estimator

Figure 2 presents the classification performance of the proposed USimUL and its corrected variant, denoted USimUL-ABS. As shown, USimUL-ABS consistently outperforms USimUL in both accuracy and stability across all datasets. These improvements demonstrate the effectiveness of corrected risk estimator in mitigating negative risks and enhancing overall performance.

Table 4: Classification accuracy under inaccurate training class priors. The true class prior $\pi_+$ is fixed at 0.40 or 0.60, while the given class prior used during training varies.

| True | Given | Fashion | Kuzushiji | CIFAR-10 |
|---|---|---|---|---|
| $\pi_+ = 0.40$ | $\pi_+ = 0.40$ | 95.51±0.04 | 87.10±0.30 | 84.62±0.31 |
| | $\pi_+ = 0.45$ | 95.40±0.05 | 86.27±0.05 | 82.84±0.09 |
| | $\pi_+ = 0.35$ | 95.40±0.03 | 86.21±0.02 | 83.41±0.15 |
| | $\pi_+ = 0.30$ | 94.99±0.17 | 86.31±0.52 | 83.47±0.29 |
| | $\pi_+ = 0.25$ | 93.34±0.29 | 85.07±0.77 | 83.24±0.33 |
| | KM1 estimated $\pi_+$ | 95.47±0.05 | 86.79±0.26 | 84.50±0.13 |
| | KM2 estimated $\pi_+$ | 95.49±0.03 | 86.63±0.52 | 84.35±0.35 |
| $\pi_+ = 0.60$ | $\pi_+ = 0.60$ | 95.78±0.06 | 88.62±0.17 | 85.22±0.06 |
| | $\pi_+ = 0.55$ | 95.76±0.13 | 88.20±0.36 | 83.74±0.19 |
| | $\pi_+ = 0.65$ | 95.71±0.09 | 88.20±0.16 | 85.03±0.26 |
| | $\pi_+ = 0.70$ | 95.43±0.05 | 87.02±0.37 | 84.96±0.28 |
| | $\pi_+ = 0.75$ | 93.53±0.27 | 86.75±0.39 | 84.74±0.13 |
| | KM1 estimated $\pi_+$ | 95.75±0.03 | 88.47±0.21 | 85.06±0.19 |
| | KM2 estimated $\pi_+$ | 95.77±0.05 | 88.41±0.39 | 84.97±0.24 |

## 4.4 Generalization across Various Class Priors

To evaluate the robustness of our method across different class priors, we conduct extensive evaluations on multiple datasets. As shown in Table 1, USimUL consistently achieves superior performance across all class priors and datasets. Specifically, as the class prior $\pi_+$ increases from 0.2 to 0.6, USimUL maintains optimal performance improvements on all datasets. Furthermore, USimUL

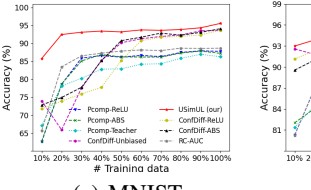

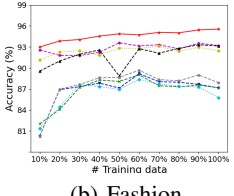

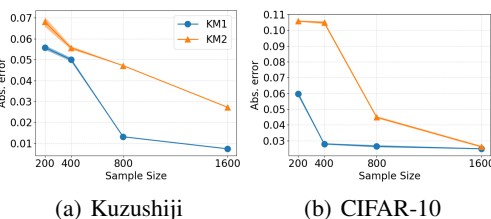

(a) MNIST        (b) Fashion        (a) Kuzushiji        (b) CIFAR-10

Figure 3: Classification accuracy of various methods when the amount of training data increases (under $\pi_+ = 0.4$).

Figure 4: Estimation errors of the class-prior (absolute value of difference between true class-prior and estimated class-prior, over 10 trails).

demonstrates greater stability, with a lower standard deviation compared to baseline and compared methods under various class priors, highlighting its strong robustness. Notably, our findings remain consistent across different types of datasets, further validating the effectiveness of our method.

### 4.5 ROBUSTNESS TO INACCURATE TRAINING CLASS PRIORS

Hitherto, we have assumed that the value of $\pi_+$ is accessible, which is rarely satisfied in practice. Fortunately, USimUL is robust to inaccurate training class priors. To demonstrate this, we set the true class prior to $\pi_+ = 0.4$ and $\pi_+ = 0.6$, and evaluate USimUL on Fashion, Kuzushiji, and CIFAR-10 using training class priors from $\{0.35, 0.45\}$ and $\{0.55, 0.65\}$. As shown in Table 4, USimUL maintains stable performance despite class prior mismatches, highlighting its robustness to inaccurate training class prior.

### 4.6 PERFORMANCE OF INCREASING TRAINING DATA

As shown in Lemma 5 and Theorem 6, the performance of our USimUL method is expected to be improved with more training data. To empirically validate this, we further conduct experiments on MNIST and Fashion with class prior $\pi_+ = 0.4$, varying the fraction of training data ($100\%$ indicates the full training data). As shown in Figure 3, the classification accuracy of USimUL generally increases as more training data become available. Its superior performance with limited data, along with its consistent accuracy improvements as training data increases, demonstrates its robustness and effectiveness. Additionally, this empirical observation aligns well with our theoretical estimation error bounds, which predict a decrease in estimation error as the amount of training data increases.

### 4.7 CLASS-PRIOR ESTIMATION

Here, we evaluate the empirical performance of class-prior estimation on datasets of sizes $\{200, 400, 800, 1600\}$, using a 1:1 ratio of uncertain similarity triplets to unlabeled samples and a true class prior of 0.4. KM1 and KM2 serve as the $CPE$ in Algorithm 2. As shown in Figure 4, the difference between the estimated class priors and the true class priors is very small. Table 4 reports the performance of the class prior estimated with KM1 and KM2, and the performance of USimUL shows no appreciable degradation. These results indicate that USimUL remains effective even without access to true class priors.

## 5 CONCLUSION

We introduce Uncertain Similarity and Unlabeled Learning (USimUL), a novel privacy-preserving framework designed to mitigate sensitive label information leakage in traditional similarity-based weakly supervised learning. USimUL introduces uncertainty components into similarity labeling. Our theoretical analysis establishes that the proposed risk estimator can reliably approximate classification risk from uncertain similarity data, achieving a statistically optimal convergence rate. Extensive experiments on benchmark and real-world datasets demonstrate that USimUL significantly outperforms existing methods.

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

## A    PROOF OF LEMMA 2.

**Lemma 2.** *Given the class priors $\pi_+ = P(y = 1)$ and $\pi_- = P(y = -1)$, and assuming that $x$, $x'$, and $x''$ are mutually independent, $P(x, x', x'', Y)$ and $P(Y)$ can be expressed as:*

$$P(x, x', x'', Y) = 2\left[\pi_+^2 P_+^2(x) + \pi_-^2 P_-^2(x)\right] P(x),$$
$$P(Y) = 1 - \pi_+\pi_-,$$
(16)

*where $P_+(x) = P(x \mid y = +1)$ and $P_-(x) = P(x \mid y = -1)$ denote the class-conditional probability densities of positive and negative samples, respectively, and $P(x)$ denotes the marginal density over all samples.*

*Proof.* Based the above definition, let $P(Y) = \{(y = y' = 1) \text{ or } (y = y' = -1) \text{ or } (y = y'' = 1) \text{ or } (y = y'' = -1)\}$. We can express $P(Y)$ as :

$$P(Y) = 1 - P(y' = y'' \neq y)$$
$$= 1 - P(y = 1, y' = y'' = -1) - P(y = -1, y' = y'' = 1).$$
(17)

Since $x$, $x'$, and $x''$ are mutually independent, we have:

$$P(Y) = 1 - P(y = 1)P(y' = -1)P(y'' = -1) - P(y = -1)P(y' = 1)P(y'' = 1)$$
$$= 1 - (\pi_+\pi_-^2) - (\pi_-\pi_+^2)$$
$$= 1 - \pi_+\pi_-(\pi_+ + \pi_-)$$
$$= 1 - \pi_+\pi_-.$$
(18)

On the other hand, the joint distribution $P(x, x', x'', Y)$ can be expanded as:

$$P(x, x', x'', Y) = P(x, x', x'', y = y' = 1) + \ldots + P(x, x', x'', y = y'' = -1)$$
$$= \pi_+ P_+(x)\pi_+ P_+(x)P(x) + \ldots + \pi_- P_-(x)\pi_- P_-(x)P(x)$$
$$= 2\left[\pi_+^2 P_+^2(x) + \pi_-^2 P_-^2(x)\right] P(x).$$
(19)

This completes the prove of Lemma 2. $\qquad\square$

## B    PROOF OF LEMMA 3.

**Lemma 3.** *The dataset $\widetilde{\mathcal{D}}_{US} \triangleq \{\widetilde{x}_i\}_{i=1}^{3N_{US}}$ consists of independently drawn samples following:*

$$\widetilde{P}_{US}(x) = \frac{2\left[\pi_+^2 P_+(x) + \pi_-^2 P_-(x)\right]}{1 - \pi_+\pi_-}.$$
(20)

*Proof.* We formally denote the set of uncertain similarity triplets as $\mathcal{D}_{US}$, defined as:

$$\mathcal{D}_{US} \triangleq \left\{\left(x_i, \{x_i', x_i''\}\right)\right\}_{i=1}^{N_{US}} \overset{i.i.d.}{\sim} P_{US}(x, \{x', x''\}),$$
(21)

where $N_{US}$ denotes the number of uncertain similarity triplets in $\mathcal{D}_{US}$. We also define the corresponding pointwise dataset $\widetilde{\mathcal{D}}_{US} \triangleq \{x_i\}_{i=1}^{3N_{US}} \overset{i.i.d.}{\sim} \widetilde{P}_{US}(x)$, which is obtained by disregarding the triplet structure in $\mathcal{D}_{US}$.

Based on Lemma 2 and Definition 1, the distribution $P_{US}(x, \{x', x''\})$ can be expressed as:

$$P_{US}(x, \{x', x''\}) = \frac{P(x, x', x'', Y)}{P(Y)}$$
$$= \frac{2}{1 - \pi_+\pi_-} \left\{\left[\pi_+^2 P_+^2(x) + \pi_-^2 P_-^2(x)\right] P(x)\right\}.$$
(22)

To derive the distribution $\widetilde{P}_{US}(x)$, we integrate both sides of Eq. (22) over $x'$ and $x''$:

$$\widetilde{P}_{US}(x) = \frac{2}{1 - \pi_+\pi_-} \left[\pi_+^2 P_+(x) \int \frac{P(x, y = 1)}{P(y = 1)} d_x + \pi_-^2 P_-(x) \int \frac{P(x, y = -1)}{P(y = -1)} d_x\right]$$
$$= \frac{2}{1 - \pi_+\pi_-} \left[\pi_+^2 P_+(x) \frac{P(y = 1)}{P(y = 1)} + \pi_-^2 P_-(x) \frac{P(y = -1)}{P(y = -1)}\right]$$
$$= \frac{2}{1 - \pi_+\pi_-} \left[\pi_+^2 P_+(x) + \pi_-^2 P_-(x)\right].$$
(23)

which concludes the proof of Lemma 3. □

## C PROOF OF THEOREM 4.

**Theorem 4.** *The classification risk can be equivalently expressed as*

$$R_{USU,\ell}(f) = \mathbb{E}_{x \sim \widetilde{P}_{US}(x)} \left\{ \bar{\ell}_+[f(x)] \right\} + \mathbb{E}_{x \sim P_U(x)} \left\{ \bar{\ell}_-[f(x)] \right\}, \tag{24}$$

*where $\bar{\ell}_+(z) = \theta_{US}^+ \ell(z, +1) + \theta_{US}^- \ell(z, -1)$ and $\bar{\ell}_-(z) = \theta_U^+ \ell(z, +1) + \theta_U^- \ell(z, -1)$.*

*Proof.* Let $\pi_+ = P(y = 1)$ and $\pi_- = P(y = -1)$ denote the class prior probabilities for the positive and negative classes, respectively. Let $P_+(x) = P(x \mid y = +1)$ and $P_-(x) = P(x \mid y = -1)$ denote the class-conditional probability densities of positive and negative samples, respectively. Under these definitions, the classification risk is given by

$$R(f) = \mathbb{E}_{P_+(x)} \pi_+ [\ell(f(x), +1)] + \mathbb{E}_{P_-(x)} \pi_- [\ell(f(x), -1)]. \tag{25}$$

On the other hand, given training data comprising uncertain similarity and unlabeled data, the classification risk can be re-expressed as:

$$\begin{aligned}
R(f) &= R_{USU,\ell}(f) \\
&= \mathbb{E}_{x \sim \widetilde{P}_{US}(x)} \left\{ \theta_{US}^+ [\ell(f(x), +1)] + \theta_{US}^- [\ell(f(x), -1)] \right\} \\
&\quad + \mathbb{E}_{x \sim P_U(x)} \left\{ \theta_U^+ [\ell(f(x), +1)] + \theta_U^- [\ell(f(x), -1)] \right\}
\end{aligned} \tag{26}$$

Using the decomposition of expectations under class priors, we have:

$$\begin{aligned}
&\mathbb{E}_{x \sim \widetilde{P}_{US}(x)} \left\{ \theta_{US}^+ [\ell(f(x), +1)] + \theta_{US}^- [\ell(f(x), -1)] \right\} \\
&= \frac{2\pi_+^2}{1 - \pi_+ \pi_-} \mathbb{E}_{x \sim \widetilde{P}_+(x)} \left\{ \theta_{US}^+ [\ell(f(x), +1)] + \theta_{US}^- [\ell(f(x), -1)] \right\} \\
&\quad + \frac{2\pi_-^2}{1 - \pi_+ \pi_-} \mathbb{E}_{x \sim \widetilde{P}_-(x)} \left\{ \theta_{US}^+ [\ell(f(x), +1)] + \theta_{US}^- [\ell(f(x), -1)] \right\},
\end{aligned} \tag{27}$$

$$\begin{aligned}
&\mathbb{E}_{x \sim P_U(x)} \left\{ \theta_U^+ [\ell(f(x), +1)] + \theta_U^- [\ell(f(x), -1)] \right\} \\
&= \pi_+ \mathbb{E}_{x \sim \widetilde{P}_+(x)} \left\{ \theta_U^+ [\ell(f(x), +1)] + \theta_U^- [\ell(f(x), -1)] \right\} \\
&\quad + \pi_- \mathbb{E}_{x \sim \widetilde{P}_-(x)} \left\{ \theta_U^+ [\ell(f(x), +1)] + \theta_U^- [\ell(f(x), -1)] \right\}.
\end{aligned} \tag{28}$$

Combining Eq. (27) and Eq. (28), we obtain

$$\begin{aligned}
R(f) &= R_{USU,\ell}(f) \\
&= \mathbb{E}_{P_+(x)} \left\{ \left[ \frac{2\pi_+^2}{1 - \pi_+ \pi_-} \theta_{US}^+ + \pi_+ \theta_U^+ \right] \ell(f(x), +1) \right. \\
&\qquad\qquad \left. + \left[ \frac{2\pi_+^2}{1 - \pi_+ \pi_-} \theta_{US}^- + \pi_+ \theta_U^- \right] \ell(f(x), -1) \right\} \\
&\quad + \mathbb{E}_{P_-(x)} \left\{ \left[ \frac{2\pi_-^2}{1 - \pi_+ \pi_-} \theta_{US}^+ + \pi_- \theta_U^+ \right] \ell(f(x), +1) \right. \\
&\qquad\qquad \left. + \left[ \frac{2\pi_-^2}{1 - \pi_+ \pi_-} \theta_{US}^- + \pi_- \theta_U^- \right] \ell(f(x), -1) \right\}.
\end{aligned} \tag{29}$$

By matching Eq. (25) and the standard classification risk in Eq. (29), we obtain

$$\begin{cases}
\frac{2\pi_+^2}{1-\pi_+\pi_-}\theta_{US}^+ + \pi_+\theta_U^+ &= \pi_+ \\
\frac{2\pi_+^2}{1-\pi_+\pi_-}\theta_{US}^- + \pi_+\theta_U^- &= 0 \\
\frac{2\pi_-^2}{1-\pi_+\pi_-}\theta_{US}^+ + \pi_-\theta_U^+ &= 0 \\
\frac{2\pi_-^2}{1-\pi_+\pi_-}\theta_{US}^- + \pi_-\theta_U^- &= \pi_-
\end{cases} \Rightarrow \begin{cases}
\theta_{US}^+ &= \frac{1-\pi_+\pi_-}{2(\pi_+-\pi_-)} \\
\theta_{US}^- &= \frac{1-\pi_+\pi_-}{2(\pi_--\pi_+)} \\
\theta_U^+ &= \frac{-2\pi_-}{2(\pi_+-\pi_-)} \\
\theta_U^- &= \frac{-2\pi_+}{2(\pi_--\pi_+)}
\end{cases} \tag{30}$$

Consequently, the classification risk is equivalently expressed as:

$$R_{USU,\ell}(f) = \mathbb{E}_{x\sim\widetilde{P}_{US}(x)}\left\{\bar{\ell}_+[f(x)]\right\} + \mathbb{E}_{x\sim P_U(x)}\left\{\bar{\ell}_-[f(x)]\right\}, \tag{31}$$

where $\bar{\ell}_+(z) = \theta_{US}^+\ell(z,+1) + \theta_{US}^-\ell(z,-1)$ and $\bar{\ell}_-(z) = \theta_U^+\ell(z,+1) + \theta_U^-\ell(z,-1)$, which completes the prove of Theorem 4. $\square$

# D    PROOF OF LEMMA 5

**Lemma 5.** *For any $\delta > 0$, with the probability at least $1 - \delta$,*

$$\sup_{\mathbf{f}\in\mathcal{F}}\left|R_{US}(\mathbf{f}) - \widehat{R}_{US}(\mathbf{f})\right| \leqslant 2L_\phi\mathfrak{R}_{N_{US}}(\mathcal{F}) + C_\phi\sqrt{\frac{2\ln(4/\delta)}{3N_{US}}}, \tag{32}$$

$$\sup_{\mathbf{f}\in\mathcal{F}}\left|R_U(\mathbf{f}) - \widehat{R}_U(\mathbf{f})\right| \leqslant 2L_\phi\mathfrak{R}_{N_U}(\mathcal{F}) + C_\phi\sqrt{\frac{2\ln(4/\delta)}{N_U}}, \tag{33}$$

*where $R_{US}(\mathbf{f}) = \mathbb{E}_{x\sim\widetilde{P}_{US}(x)}\bar{\ell}_+[f(x)]$, $R_U(\mathbf{f}) = \mathbb{E}_{x\sim P_U(x)}\bar{\ell}_-[f(x)]$, and $\widehat{R}_{US}(\mathbf{f})$ and $\widehat{R}_U(\mathbf{f})$ denote the empirical risk estimator to $R_{US}(\mathbf{f})$ and $R_U(\mathbf{f})$, respectively. $\mathfrak{R}_{N_{US}}(\mathcal{F})$, and $\mathfrak{R}_{N_U}(\mathcal{F})$ are the Rademacher complexities(Mohri et al., 2018) of $\mathcal{F}$ for the sampling of size $3N_{US}$ from $\widetilde{P}_{US}(x)$ and the sampling of size $N_U$ from $P_U(x)$.*

*Proof.* Since the surrogate loss $\phi(z)$ is bounded by $\sup_z\phi(z) \leqslant C_\phi$, let function $\Phi_{US}$ defined for any uncertain similarity samples $S_{US}$ by $\Phi(S_{US}) = \sup_{\mathbf{f}\in\mathcal{F}}R_{US}(\mathbf{f}) - \widehat{R}_{US}(\mathbf{f})$. If $x_i$ in unconcealed labels dataset is replaced with $x_i'$, the change of $\Phi_{US}(S_US)$ does not exceed the supermum of the difference, we have

$$\Phi_{US}(S_{US}') - \Phi_{US}(S_US) \leqslant \frac{2C_\phi}{3N_{US}} \tag{34}$$

Then, by McDiarmid's inequality, for any $\delta > 0$, with probability at least $1 - \delta$, the following holds:

$$\sup_{\mathbf{f}\in\mathcal{F}}|\widehat{R}_{US}(\mathbf{f}) - R_{US}(\mathbf{f})| \leqslant \mathbb{E}\left[\Phi_{US}(S_{US})\right] + C_\phi\sqrt{\frac{2\ln(4/\delta)}{3N_{US}}}. \tag{35}$$

Hence, by using the Rademacher complexity (Mohri et al., 2018), we can obtain

$$\sup_{\mathbf{f}\in\mathcal{F}}|\widehat{R}_{US}(\mathbf{f}) - R_{US}(\mathbf{f})| \leqslant 2\mathfrak{R}_{N_{US}}(\widetilde{l}_{US}\circ\mathcal{F}) + C_\phi\sqrt{\frac{2\ln(4/\delta)}{3N_{US}}}, \tag{36}$$

where $\mathfrak{R}_{N_{US}}(\widetilde{l}_{US}\circ\mathcal{F})$ is the Rademacher complexity of the composite function class $(\widetilde{l}_{US}\circ\mathcal{F})$ for examples size $N_{US}$. As $L_\phi$ is the Lipschitz constant of $\phi$, we have $\mathfrak{R}_{N_{US}}(\widetilde{l}_{US}\circ\mathcal{F}) \leqslant L_\phi\mathfrak{R}_{N_{US}}(\mathcal{F})$ by Talagrand's contraction Lemma (Mohri et al., 2018). Then, we can obtain the

$$\sup_{\mathbf{f}\in\mathcal{F}}\left|R_{US}(\mathbf{f}) - \widehat{R}_{US}(\mathbf{f})\right| \leqslant 2L_\phi\mathfrak{R}_{N_{US}}(\mathcal{F}) + C_\phi\sqrt{\frac{2\ln(4/\delta)}{3N_{US}}} \tag{37}$$

Then, $\sup_{\mathbf{f}\in\mathcal{F}}\left|R_U(\mathbf{f}) - \widehat{R}_U(\mathbf{f})\right|$ can be proven using the same proof technique, which finishes the proof of Lemma 5. $\square$

# E    PROOF OF THEOREM 6.

**Theorem 6.** *For any $\delta > 0$, with the probability at least $1 - \delta$,*

$$R_{USU}(\hat{\mathbf{f}}) - \min_{\mathbf{f}\in\mathcal{F}}R_{USU}(\mathbf{f}) \leqslant 4L_\phi\mathfrak{R}_{N_{US}}(\mathcal{F}) + 4L_\phi\mathfrak{R}_{N_U}(\mathcal{F})$$

$$+ 2C_\phi\sqrt{\frac{2\ln(4/\delta)}{3N_{US}}} + 2C_\phi\sqrt{\frac{2\ln(4/\delta)}{N_U}}, \tag{38}$$

*where $\widehat{\mathbf{f}}$ is trained by minimizing the classification risk $R_{USU}$.*

*Proof.* According to Lemma 4, the estimation error bound is proven through

$$R_{USU}(\widehat{\mathbf{f}}_{USU}) - R_{USU}(\mathbf{f}^*) = (\widehat{R}_{USU}(\widehat{\mathbf{f}}_{USU}) - \widehat{R}_{USU}(\widehat{\mathbf{f}}^*)) + (R(\widehat{\mathbf{f}}_{USU}) - \widehat{R}_{USU}(\widehat{\mathbf{f}}_{USU}))$$
$$+ (\widehat{R}_{USU}(\widehat{\mathbf{f}}^*) - R(\widehat{\mathbf{f}}^*)) \tag{39}$$
$$\leqslant 0 + 2 sup_{\mathbf{f} \in \mathcal{F}} |R_{USU}(\mathbf{f}) - \widehat{R}_{USU}(\mathbf{f})|$$

where $\mathbf{f}^* = arg\min_{\mathbf{f} \in \mathcal{F}} R(\mathbf{f})$.

Now, we have seen the definition of $R_{USU}(\mathbf{f})$ and $\widehat{R}_{USU}(\mathbf{f})$, which can also be decomposed into:

$$R_{USU}(f) = \mathbb{E}_{x \sim \widetilde{P}_{US}(x)} \left\{ \bar{\ell}_+[f(x)] \right\} + \mathbb{E}_{x \sim P_U(x)} \left\{ \bar{\ell}_-[f(x)] \right\}, \tag{40}$$

and

$$\widehat{R}_{USU}(f) = \frac{1}{3N_{US}} \sum_{i=1}^{3N_{US}} \left\{ \bar{\ell}_+[f(x_i)] \right\} + \frac{1}{N_U} \sum_{j=1}^{N_U} \left\{ \bar{\ell}_-[f(x_j)] \right\}. \tag{41}$$

Due to the sub-additivity of the supremum operators, it holds that

$$sup_{\mathbf{f} \in \mathcal{F}} |\widehat{R}_{USU}(\mathbf{f}) - R_{USU}(\mathbf{f})| \leqslant sup_{\mathbf{f} \in \mathcal{F}} |\widehat{R}_{US}(\mathbf{f}) - R_{US}(\mathbf{f})|$$
$$+ sup_{\mathbf{f} \in \mathcal{F}} |\widehat{R}_U(\mathbf{f}) - R_U(\mathbf{f})| \tag{42}$$

where

$$R_{US}(\mathbf{f}) = \mathbb{E}_{x \sim \widetilde{P}_{US}(x)} \left\{ \bar{\ell}_+[f(x)] \right\}$$
$$\widehat{R}_{US}(\mathbf{f}) = \frac{1}{3N_{US}} \sum_{i=1}^{3N_{US}} \left\{ \bar{\ell}_+[f(x_i)] \right\}$$
$$R_U(\mathbf{f}) = \mathbb{E}_{x \sim P_U(x)} \left\{ \bar{\ell}_-[f(x)] \right\} \tag{43}$$
$$\widehat{R}_U(\mathbf{f}) = \frac{1}{N_U} \sum_{j=1}^{N_U} \left\{ \bar{\ell}_-[f(x_j)] \right\}.$$

According to the Lemma 5, we can get the generalization bound that

$$R_{USU}(\widehat{\mathbf{f}}) - \min_{\mathbf{f} \in \mathcal{F}} R_{USU}(\mathbf{f}) \leqslant 4L_\phi \mathfrak{R}_{N_{US}}(\mathcal{F}) + 4L_\phi \mathfrak{R}_{N_U}(\mathcal{F})$$
$$+ 2C_\phi \sqrt{\frac{2\ln(4/\delta)}{3N_{US}}} + 2C_\phi \sqrt{\frac{2\ln(4/\delta)}{N_U}} \tag{44}$$

with probability at least $1 - \delta$, which finishes the proof of Theorem 6. □

# F ADDITIONAL EXPERIMENTS.

To supplement the main text, this section presents additional experimental results and analyses, including further validation on real-world datasets, an investigation into the impact of increased unlabeled data, a discussion on training convergence, extended results on additional UCI datasets, and extended results on inaccurate training class prior.

## F.1 FURTHER EVALUATION ON REAL-WORLD WSL DATASETS

We further evaluate our method on additional real-world datasets, including Pendigits, Lost, MSRCv2, BirdSong, and Yahoo! News, with the class prior $\pi_+ = 0.6$. The results are summarized in Table 5. As observed, USimUL consistently outperforms all baseline and comparison methods across these datasets, demonstrating strong overall performance. Moreover, USimUL generally achieves lower standard deviations, highlighting its robustness and stability. These results provide further empirical evidence of the effectiveness and reliability of our approach.

Table 5: Classification accuracy of each algorithm on real-world WSL datasets. We report the mean and standard deviation of results over 5 trials. The best method is highlighted in **bold** and the second-best method is underlined (under 5% t-test, $\pi_+ = 0.6$).

| Setting | Method | Pendigits | Lost | MSRCv2 | BirdSong | Yahoo! News |
|---------|--------|-----------|------|--------|----------|-------------|
| Baselines | Sconf-ABS | $75.15 \pm 0.54$ | $65.36 \pm 0.88$ | $62.18 \pm 0.52$ | $67.00 \pm 2.09$ | $61.79 \pm 0.47$ |
| | Sconf-NN | $78.31 \pm 0.77$ | $66.56 \pm 0.31$ | $69.43 \pm 0.57$ | $67.94 \pm 0.81$ | $61.91 \pm 0.16$ |
| Conf Comparison | Pcomp-ReLU | $89.02 \pm 0.62$ | $74.38 \pm 1.88$ | $74.86 \pm 1.29$ | $73.73 \pm 0.66$ | $73.11 \pm 1.20$ |
| | Pcomp-ABS | $85.66 \pm 0.19$ | $72.81 \pm 0.31$ | $70.47 \pm 1.04$ | $73.95 \pm 0.66$ | $69.51 \pm 1.52$ |
| | Pcomp-Teacher | $90.33 \pm 0.79$ | $73.68 \pm 0.72$ | $74.77 \pm 0.49$ | $74.61 \pm 0.22$ | $73.97 \pm 0.64$ |
| | PC-AUC | $88.57 \pm 0.25$ | $73.97 \pm 2.10$ | $72.79 \pm 0.26$ | $76.82 \pm 0.66$ | $72.78 \pm 0.67$ |
| Conf Difference | ConfDiff-Unbiased | $93.11 \pm 0.44$ | $68.09 \pm 1.23$ | $72.20 \pm 0.52$ | $76.38 \pm 1.55$ | $72.59 \pm 0.39$ |
| | ConfDiff-ReLU | $93.97 \pm 0.35$ | $67.34 \pm 0.91$ | $75.24 \pm 0.97$ | $78.46 \pm 0.53$ | $72.18 \pm 1.03$ |
| | ConfDiff-ABS | $94.55 \pm 0.17$ | $66.35 \pm 0.10$ | $74.63 \pm 2.05$ | $78.87 \pm 0.72$ | $73.36 \pm 1.17$ |
| | **USimUL (Our)** | $\mathbf{97.22 \pm 0.25}$ | $\mathbf{78.95 \pm 1.32}$ | $\mathbf{79.02 \pm 2.85}$ | $\mathbf{82.45 \pm 0.33}$ | $\mathbf{75.83 \pm 1.48}$ |

Table 6: Classification accuracy of each algorithm on real-world privacy-sensitive datasets (under 5% t-test, $\pi_+ = 0.6$). The best method is highlighted in **bold** and the second-best method is underlined .

| Setting | Method | DDSM | PDMD | PDSD |
|---------|--------|------|------|------|
| Baselines | Sconf-ABS | $63.06 \pm 0.11$ | $83.25 \pm 2.36$ | $68.13 \pm 0.63$ |
| | Sconf-NN | $62.57 \pm 1.40$ | $84.38 \pm 3.12$ | $67.75 \pm 1.25$ |
| Conf Comparison | Pcomp-ReLU | $\mathbf{78.38 \pm 0.37}$ | $87.37 \pm 2.95$ | $76.66 \pm 3.11$ |
| | Pcomp-ABS | $72.94 \pm 1.25$ | $83.94 \pm 2.49$ | $74.99 \pm 1.80$ |
| | Pcomp-Teacher | $69.82 \pm 1.61$ | $85.85 \pm 3.37$ | $75.84 \pm 1.12$ |
| | PC-AUC | $69.52 \pm 0.34$ | $77.95 \pm 6.52$ | $67.50 \pm 2.04$ |
| Conf Difference | ConfDiff-Unbiased | $76.13 \pm 0.81$ | $91.75 \pm 0.54$ | $74.11 \pm 2.57$ |
| | ConfDiff-ReLU | $72.36 \pm 1.42$ | $87.77 \pm 3.41$ | $71.57 \pm 2.03$ |
| | ConfDiff-ABS | $74.02 \pm 0.65$ | $91.28 \pm 0.38$ | $73.60 \pm 2.82$ |
| | **USimUL (Our)** | $76.33 \pm 0.14$ | $\mathbf{95.83 \pm 0.04}$ | $\mathbf{84.38 \pm 0.62}$ |

## F.2 FURTHER EVALUATION ON REAL-WORLD PRIVACY-SENSITIVE DATASETS

We further evaluate our method on additional real-world privacy-sensitive datasets, including DDSM, PDMD, and PDSD, with the class prior $\pi_+ = 0.6$. The results are summarized in Table 6. As observed, USimUL shows consistent improvement over baselines and comparison methods across these datasets, demonstrating strong overall performance.

## F.3 IMPACT OF UNLABELED DATA QUANTITY

To evaluate the impact of increasing the amount of unlabeled data, we conduct additional ablation experiments on MNIST and Fashion-MNIST with class priors $\pi_+ = 0.4$ and $\pi_+ = 0.6$. As shown in Figure 5, USimUL consistently achieves the highest accuracy across all levels of unlabeled data. In contrast, certain baselines, such as Pcomp-ReLU and Pcomp-Teacher, exhibit limited improvement, indicating their inefficacy in utilizing additional unlabeled information. These results further underscore USimUL's superior capability in leveraging unlabeled data for performance enhancement, reinforcing its robustness in weakly supervised learning scenarios.

## F.4 CONVERGENCE SPEED ANALYSIS

Figure 6 and Figure 7 show how quickly our model converges. As illustrated, our method (represented by the red solid line) reaches convergence at around 20 epochs. This rapid convergence demonstrates the efficiency and stability of our method. It also suggests that our method can achieve

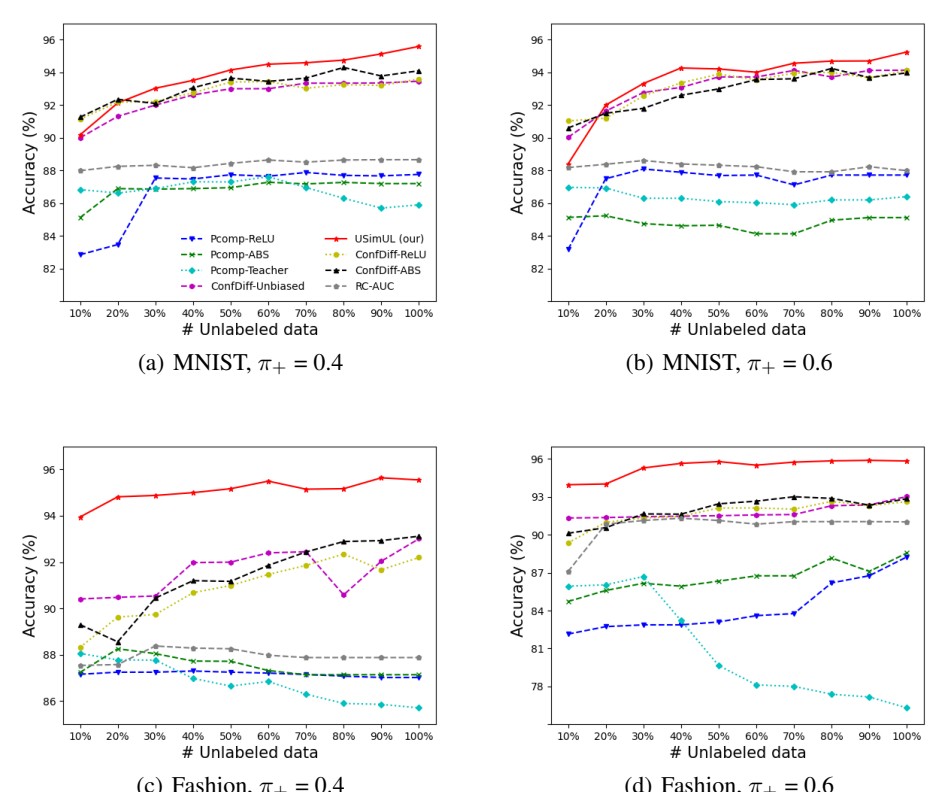

(a) MNIST, $\pi_+ = 0.4$  (b) MNIST, $\pi_+ = 0.6$

(c) Fashion, $\pi_+ = 0.4$  (d) Fashion, $\pi_+ = 0.6$

Figure 5: The classification accuracy of various methods when the amount of unlabeled data increases.

Table 7: Classification accuracy of each algorithm on Yahoo! News datasets (under 5% t-test, $\pi_+ = 0.6$). The best method is highlighted in **bold** and the second-best method is underlined.

| Setting | Method | Yahoo! News |
|---|---|---|
| Baselines | Sconf-ABS | $60.09 \pm 0.07$ |
| | Sconf-NN | $60.54 \pm 0.32$ |
| Conf Comparison | Pcomp-ReLU | $74.48 \pm 0.89$ |
| | Pcomp-ABS | $68.69 \pm 0.73$ |
| | Pcomp-Teacher | $75.58 \pm 0.33$ |
| | PC-AUC | $75.64 \pm 1.35$ |
| Conf Difference | ConfDiff-Unbiased | $74.34 \pm 0.22$ |
| | ConfDiff-ReLU | $73.79 \pm 0.93$ |
| | ConfDiff-ABS | $75.13 \pm 1.64$ |
| | **USimUL (Our)** | $79.75 \pm 0.34$ |

strong performance with fewer training iterations, which is particularly advantageous in scenarios with limited computational resources or time constraints.

Table 8: Classification accuracy of given inaccurate training class priors.

| True | Given | MNIST | SVHN |
|---|---|---|---|
| $\pi_+ = 0.40$ | $\pi_+ = 0.35$ | 94.99±0.14 | 87.21±0.21 |
| | $\pi_+ = 0.45$ | 95.28±0.18 | 87.44±1.11 |
| | $\pi_+ = 0.40$ | 95.36±0.23 | 87.18±0.95 |
| $\pi_+ = 0.60$ | $\pi_+ = 0.55$ | 94.67±0.10 | 86.92±0.08 |
| | $\pi_+ = 0.65$ | 95.00±0.02 | 87.60±0.34 |
| | $\pi_+ = 0.60$ | 95.05±0.20 | 87.92±0.12 |

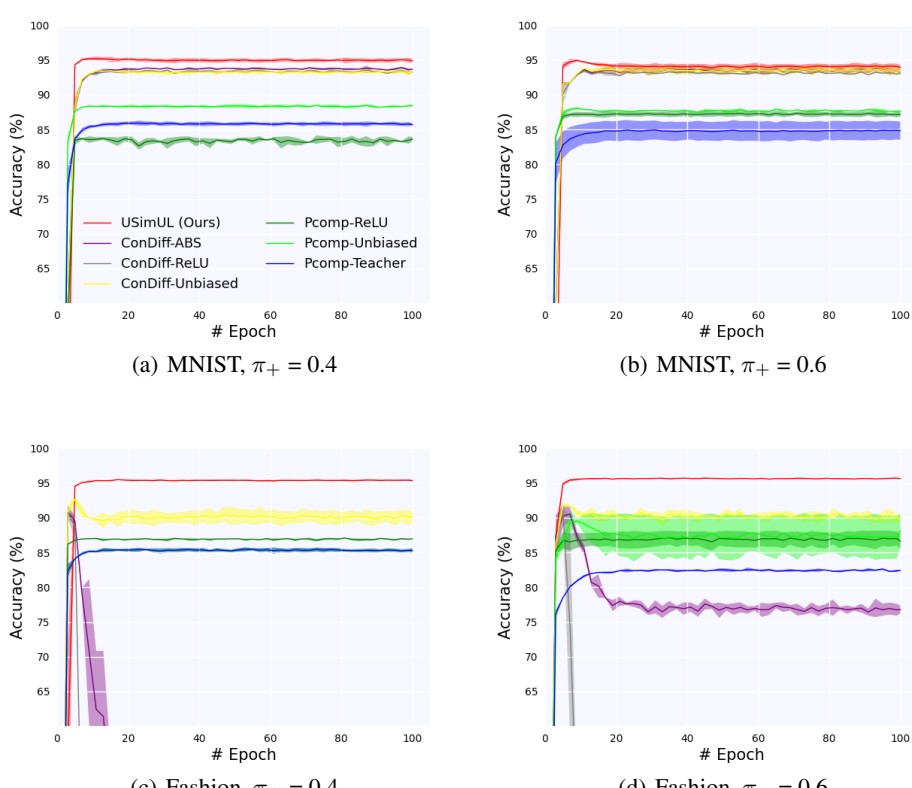

(a) MNIST, $\pi_+ = 0.4$  (b) MNIST, $\pi_+ = 0.6$

(c) Fashion, $\pi_+ = 0.4$  (d) Fashion, $\pi_+ = 0.6$

Figure 6: Experimental results on MNIST and Fashion datasets with varying class priors.

## F.5 EXTENDED RESULTS WITH INACCURATE TRAINING CLASS PRIOR

Table 8 presents the extended results with inaccurate training class prior. We set the true class prior to $\pi_+ = 0.4$ and $\pi_+ = 0.6$, and evaluate USimUL on MNIST and SVHN datasets using training class priors from $\{0.35, 0.45\}$ and $\{0.55, 0.65\}$. As shown in Table 8, USimUL maintains stable performance despite class prior mismatches, highlighting its robustness to inaccurate training class prior.

## F.6 EMPIRICAL RESULTS OF PRIVACY PROTECTION

To further demonstrate the effectiveness of the proposed USimUL in privacy protection, we add an experiment in which an attacker attempts to directly infer hidden labels using known partial information. We report how different proportions of sample leakage affect the number of exposed samples and the protection efficiency of each method, under deterministic pairs (Baselines) and

Table 9: Empirical results of privacy protection efficiency on CIFAR-10 (5 trials).

| Init Leak Ratio | Scenario | #Init Leaked | #Similar at Risk (Baseline) | #Actually Leaked (Baseline) | #Actually Leaked (Ours) | Baseline Leak Rate (%) | Ours Leak Rate (%) | Ours Protected (%) |
|---|---|---|---|---|---|---|---|---|
| **10%** | (1) A→B | 600 | 600 | 600 | 304.38±5.88 | 100 | 50.73±0.98 | 49.27±0.98 |
| | (2) A→C | 600 | 600 | 600 | 303.42±8.04 | 100 | 50.57±1.34 | 49.43±1.34 |
| | (3) B→A | 600 | 600 | 600 | 304.02±12.54 | 100 | 50.67±2.09 | 49.33±2.10 |
| | (4) C→A | 600 | 600 | 600 | 303.18±5.82 | 100 | 50.53±0.97 | 49.47±0.97 |
| | (5) B&C→A | 1200 | 1200 | 1200 | 297.18±7.86 | 100 | 24.77±0.66 | 75.23±0.66 |
| **20%** | (1) A→B | 1200 | 1200 | 1200 | 596.40±7.56 | 100 | 49.70±0.63 | 50.30±0.63 |
| | (2) A→C | 1200 | 1200 | 1200 | 589.08±9.36 | 100 | 49.09±0.78 | 50.91±0.48 |
| | (3) B→A | 1200 | 1200 | 1200 | 599.40±20.04 | 100 | 49.95±1.67 | 50.05±1.67 |
| | (4) C→A | 1200 | 1200 | 1200 | 589.20±10.44 | 100 | 49.10±0.87 | 50.90±0.87 |
| | (5) B&C→A | 2400 | 2400 | 2400 | 611.64±15.60 | 100 | 25.49±0.65 | 74.51±0.65 |
| **30%** | (1) A→B | 1800 | 1800 | 1800 | 884.88±18.36 | 100 | 49.16±1.02 | 50.84±1.02 |
| | (2) A→C | 1800 | 1800 | 1800 | 898.74±16.38 | 100 | 49.93±0.91 | 50.07±0.91 |
| | (3) B→A | 1800 | 1800 | 1800 | 902.52±7.20 | 100 | 50.14±0.40 | 49.86±0.40 |
| | (4) C→A | 1800 | 1800 | 1800 | 902.34±19.44 | 100 | 50.13±1.08 | 49.87±1.08 |
| | (5) B&C→A | 3600 | 3600 | 3600 | 888.12±21.78 | 100 | 24.67±0.61 | 75.33±0.61 |

Table 10: Empirical results of privacy protection efficiency on SVHN (5 trials).

| Init Leak Ratio | Scenario | #Init Leaked | #Similar at Risk (Baseline) | #Actually Leaked (Baseline) | #Actually Leaked (Ours) | Baseline Leak Rate (%) | Ours Leak Rate (%) | Ours Protected (%) |
|---|---|---|---|---|---|---|---|---|
| **10%** | (1) A→B | 800 | 800 | 800 | 405.60±13.68 | 100 | 50.70±1.71 | 49.30±1.71 |
| | (2) A→C | 800 | 800 | 800 | 403.52±9.04 | 100 | 50.44±1.13 | 49.56±1.13 |
| | (3) B→A | 800 | 800 | 800 | 404.64±10.32 | 100 | 50.58±1.29 | 49.42±1.29 |
| | (4) C→A | 800 | 800 | 800 | 397.20±22.24 | 100 | 49.65±2.78 | 50.35±2.78 |
| | (5) B&C→A | 1600 | 1600 | 1600 | 405.04±15.04 | 100 | 25.32±0.94 | 74.68±0.94 |
| **20%** | (1) A→B | 1600 | 1600 | 1600 | 811.04±23.68 | 100 | 50.69±1.48 | 49.31±1.48 |
| | (2) A→C | 1600 | 1600 | 1600 | 804.00±13.76 | 100 | 50.25±0.86 | 49.75±0.86 |
| | (3) B→A | 1600 | 1600 | 1600 | 795.04±16.32 | 100 | 49.69±1.02 | 50.31±1.02 |
| | (4) C→A | 1600 | 1600 | 1600 | 792.48±15.68 | 100 | 49.53±0.98 | 50.47±0.98 |
| | (5) B&C→A | 3200 | 3200 | 3200 | 795.04±16.16 | 100 | 24.85±0.51 | 75.15±0.51 |
| **30%** | (1) A→B | 2400 | 2400 | 2400 | 1203.12±21.12 | 100 | 50.13±0.88 | 49.87±0.88 |
| | (2) A→C | 2400 | 2400 | 2400 | 1195.92±19.44 | 100 | 49.83±0.81 | 50.17±0.81 |
| | (3) B→A | 2400 | 2400 | 2400 | 1214.16±15.36 | 100 | 50.59±0.64 | 49.41±0.64 |
| | (4) C→A | 2400 | 2400 | 2400 | 1180.32±18.00 | 100 | 49.18±0.75 | 50.82±0.75 |
| | (5) B&C→A | 4800 | 4800 | 4800 | 1205.52±26.64 | 100 | 25.12±0.56 | 74.88±0.56 |

uncertain similarity triplets (Ours). Specifically, for the triplet dataset (A, B, C), we simulate five attack models as follows:

(1) A→B: Given 10%, 20%, and 30% label leakage of sample A, the attacker infers the label of the bound sample B based on the binding relationship. a) In the traditional pairwise similarity structure, this binding directly exposes B's label, so the inference success rate for sample B is 100% (Baseline Leak Rate). b) In the uncertain similarity triplet, since it's unclear whether B or C is similar to A, the attacker assumes A is bound to B and assigns A's label to B. The inference success rate for B is then recorded (Success Infer Rate).

(2) A→C: Given 10%, 20%, and 30% label leakage of sample A, the attacker infers the label of the bound sample C. a) In the traditional pairwise structure, the binding directly exposes C's label, so the inference success rate for C is 100% (Baseline Leak Rate). b) In the uncertain similarity triplet, the attacker assigns A's label to C based on the assumption that A is bound to C. The inference success rate for C is then recorded (Success Infer Rate).

(3) B→A: Given 10%, 20%, and 30% label leakage of sample B, the attacker infers the label of the bound sample A. a) In the traditional pairwise similarity structure, this binding directly exposes A's label, so the inference success rate for sample A is 100% (Baseline Leak Rate). b) In the uncertain similarity triplet, since it's unclear whether B or C is similar to A, the attacker assumes B is bound to A and assigns B's label to A. The inference success rate for A is then recorded (Success Infer Rate).

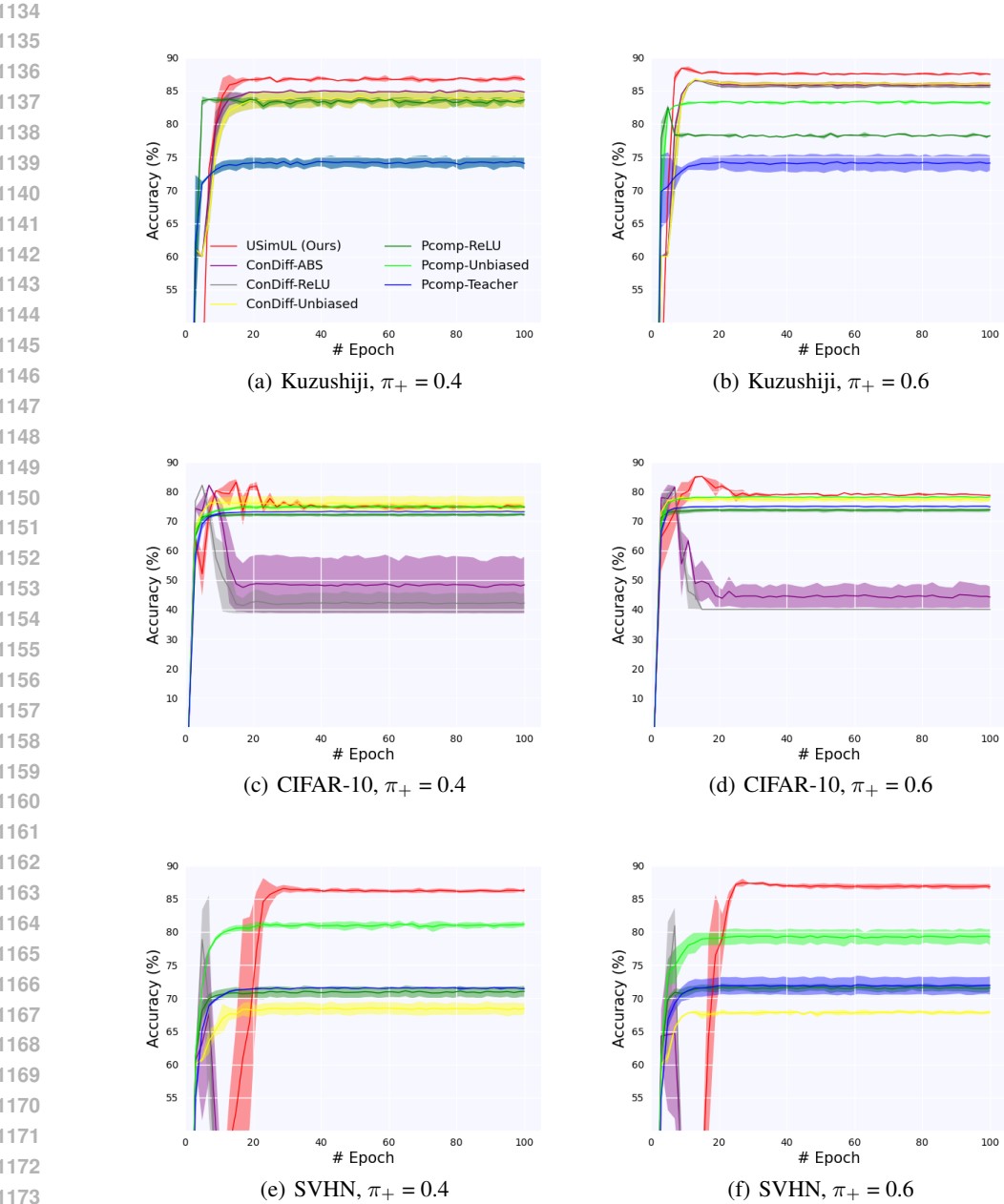

Figure 7: Experimental results on Kuzushiji, CIFAR-10 and SVHN datasets with varying class priors.

(4) C→A: Given 10%, 20%, and 30% label leakage of sample C, the attacker infers the label of the bound sample A. a) In the traditional pairwise structure, this binding directly exposes A's label, so the inference success rate for sample A is 100% (Baseline Leak Rate). b) In the uncertain similarity triplet, since it's unclear whether B or C is similar to A, the attacker assumes C is bound to A and assigns C's label to A. The inference success rate for A is then recorded (Success Infer Rate).

(5) B & C→A: Given 10%, 20%, and 30% label leakage of samples B & C, the attacker infers the label of the bound sample A based on the binding relationship between B and C. a) In the traditional pairwise similarity structure, this binding directly exposes A's label, so the inference success rate for sample A is 100% (Baseline Leak Rate). b) In the uncertain similarity triplet, since it's unclear whether B or C is similar to A, the attacker assigns labels as follows: when B and C labels are the

same, both labels are assigned to A; when B and C labels differ, one label is randomly assigned to A. The inference success rate for A is then recorded (Success Infer Rate).

Finally, we report the number of successfully inferred (i.e., exposed) samples. Preliminary tests on CIFAR-10 (see Table 9) and SVHN (see Table 10) show that label inference accuracy drops from 100% to ≈50% (results are averaged over five independent trials). It is nearly equivalent to random guessing in the context of a binary classification task, which demonstrates that USimUL can effectively alleviate the privacy risk in similarity-based weakly supervised learning.

### F.7 DETAILS OF CLASS-PRIOR ESTIMATION RESULTS

Here, we present the detail results of class-prior estimation on five benchmark datasets. We use KM1 and KM2 as the $CPE$ in the Algorithm 2. For each true class prior (0.4 and 0.6), we report the mean and standard deviation of the absolute difference between the true class prior and the estimated class priors from the training data over 10 trials. As shown in Table 11 and Table 12, as the sample size increases to 1600, the estimation error can be reduced to below 0.03. We believe that with further increases in the sample size, this estimation error can be further minimized. These results demonstrate that our method is feasible even without relying on the true priors.

Table 11: Class-prior estimation results (mean $\pm$ std) across different sample sizes with the true class prior $\pi_+$ set to 0.4 (over 10 trials).

| Dataset | | 200 | 400 | 800 | 1600 |
|---|---|---|---|---|---|
| MNIST | KM1 | 0.08413 ± 0.00130 | 0.08603 ± 0.00096 | 0.03578 ± 0.00012 | 0.01092 ± 0.00002 |
| | KM2 | 0.02247 ± 0.00021 | 0.01541 ± 0.00009 | 0.00981 ± 0.00002 | 0.00611 ± 0.00003 |
| Kuzushiji | KM1 | 0.05588 ± 0.00104 | 0.05006 ± 0.00085 | 0.01311 ± 0.00004 | 0.00735 ± 0.00009 |
| | KM2 | 0.06808 ± 0.00233 | 0.05571 ± 0.00076 | 0.04717 ± 0.00010 | 0.02724 ± 0.00018 |
| CIFAR-10 | KM1 | 0.05969 ± 0.00126 | 0.02789 ± 0.00045 | 0.02640 ± 0.00070 | 0.02481 ± 0.00025 |
| | KM2 | 0.10575 ± 0.00033 | 0.10483 ± 0.00091 | 0.04482 ± 0.00064 | 0.02614 ± 0.00055 |
| SVHN | KM1 | 0.07379 ± 0.00389 | 0.06310 ± 0.00135 | 0.04523 ± 0.00063 | 0.03940 ± 0.00049 |
| | KM2 | 0.03007 ± 0.00069 | 0.03183 ± 0.00052 | 0.02286 ± 0.00024 | 0.01349 ± 0.00016 |

Table 12: Class-prior estimation results (mean $\pm$ std) across different sample sizes with the true class prior $\pi_+$ set to 0.6 (over 10 trials.)

| Dataset | | 200 | 400 | 800 | 1600 |
|---|---|---|---|---|---|
| MNIST | KM1 | 0.09783 ± 0.01656 | 0.04445 ± 0.00297 | 0.04008 ± 0.00019 | 0.01940 ± 0.00008 |
| | KM2 | 0.03901 ± 0.00119 | 0.02761 ± 0.00023 | 0.02417 ± 0.00009 | 0.01846 ± 0.00002 |
| Kuzushiji | KM1 | 0.08990 ± 0.02207 | 0.06647 ± 0.00938 | 0.02495 ± 0.00028 | 0.01193 ± 0.00002 |
| | KM2 | 0.09346 ± 0.01640 | 0.07480 ± 0.01111 | 0.04046 ± 0.00070 | 0.02096 ± 0.00003 |
| CIFAR-10 | KM1 | 0.07533 ± 0.02013 | 0.04836 ± 0.00020 | 0.01855 ± 0.00015 | 0.01521 ± 0.00008 |
| | KM2 | 0.09489 ± 0.03226 | 0.04675 ± 0.00006 | 0.03498 ± 0.00013 | 0.02191 ± 0.00004 |
| SVHN | KM1 | 0.09266 ± 0.00176 | 0.07027 ± 0.00122 | 0.05199 ± 0.00048 | 0.04034 ± 0.00028 |
| | KM2 | 0.01268 ± 0.00004 | 0.00991 ± 0.00004 | 0.01081 ± 0.00003 | 0.00772 ± 0.00003 |

## G DETAILS OF DATASETS.

The summary statistics of four benckmark datasets and the sources of these datasets are as follows:

- MNIST (LeCun et al., 1998): The MNIST dataset is a handwritten digits dataset, which is composed of 10 classes. Each sample is a $28 \times 28$ grayscale image. The MNIST dataset has 60k training examples and 10k test examples. Source: http://yann.lecun.com/exdb/mnist/
- Fashion (Xiao et al., 2017): The Fashion dataset for classifying fashion consists of pictures from 10 classes: t-shirt, trouser, pillover, dress, coat, sandal, shirt, sneaker, bag, ankle boot. The training dataset has 6,000 images for each class, and the test dataset contains 1,000 images. Each input image is 28 pixels wide and high. Source: https://github.com/zalandoresearch/fashion-mnist

---

**Algorithm 1** Learning from Uncertain Similarity and Unlabeled Data

---

**Input:**

$\mathcal{D}_{US} = \left\{ \left( x_i, \{x_i', x_i''\} \right) \right\}_{i=1}^{N_{US}}$ and $\mathcal{D}_U = \{x_i\}_{i=1}^{N_U}$ are sampled independently from $P_{US}(x, \{x', x''\})$ and $P_U(x)$;

The number of epochs $T$;

The number of batches $B$;

**for** $t = 1$ to $T$ **do**

    Obtain $\widetilde{\mathcal{D}}_{US} = \{x_i\}_{i=1}^{3N_{US}}$ by disassembling $\mathcal{D}_{US}$;

    Obtain $\mathcal{D} = \{x_i\}_{i=1}^{3N_{US}+N_U}$ by merging $\widetilde{\mathcal{D}}_{US}$ and $\mathcal{D}_U$;

    Shuffle training set $\mathcal{D}$ into $B$ mini-batches;

    **for** $b = 1$ to $B$ **do**

        **Calculate** $\bar{\ell}_+[f(x_i)]$ and $\bar{\ell}_-[f(x_i)]$;

        **Update** model parameters $\theta$ by $\widehat{R}_{USU,\ell}(f)$ in Eq. (10) in the main manuscript;

    **end for**

**end for**

**Output:** Model parameter $\theta$ for $f(\boldsymbol{x}, \theta)$;

---

---

**Algorithm 2** Class-prior estimation from USimU data.

---

**Input:**

$\mathcal{D}_U = \{x_i\}_{i=1}^{N_U}$ (samples from $P$), $\tilde{\mathcal{D}}_{US} = \{x_i\}_{i=1}^{3N_{US}}$ (samples from $\tilde{P}_{US}$).

$CPE(,)$ is a class prior estimation algorithm.

**Output:** class prior $\pi_+$

$\pi_{US} \leftarrow CPE(\mathcal{D}, \tilde{\mathcal{D}}_{US})$

$\pi_+ \leftarrow \frac{\sqrt{4\pi_{US}-3}+1}{2}$

---

- Kuzushiji (Clanuwat et al., 2018): Similar to MNIST, Kuzushiji contains 60k training examples and 10k test examples from 10 classes. Each sample is a $28 \times 28$ grayscale image. Source: `https://github.com/rois-codh/kmnist`

- CIFAR-10 (Torralba et al., 2008): The CIFAR-10 dataset has 10 classes of various objects: airplane, automobile, bird, cat, etc. This dataset has 50k training samples and 10k test samples and each sample is a colored image in $32 \times 32 \times 3$ RGB formats. Source: `https://www.cs.toronto.edu/~kriz/cifar.html`

- SVHN (Netzer et al., 2011) : The SVHN dataset is a street view house number dataset, which is composed of 10 classes. Each sample is a $32 \times 32 \times 3$ RGB image. This dataset has 73,257 training examples and 26,032 test examples. Source: `http://ufldl.stanford.edu/housenumbers/`

Table 14 provides a summary of all datasets used, along with their corresponding base models.

## H  STEP-BY-STEP ALGORITHM

To help non-expert readers better understand the procedure, we present a step-by-step algorithm in Algorithm 1.

## I  COMPARISON WITH BASELINES IN PRIVACY PROTECTION EFFECTIVENESS

We present a comparison with baselines in privacy protection effectiveness in Table 13.

## J  LIMITATION AND FUTURE WORK.

While USimUL effectively balances privacy protection and model performance, its current design primarily targets binary classification. In fact, our method can be extended to multi-class clas-

Table 13: Comparison with Baselines in Privacy Protection Effectiveness

| Methods | Data | Label | Privacy protection effectiveness | if $x_1$ is exposed |
|---|---|---|---|---|
| Similarity-pairs | $(x_1, x_2)$ | $y_1 = y_2$ | No privacy protection | $x_2$ will be exposed |
| Similarity-Conf | $(x_1, x_2, s)$ | $s = sim(y_1, y_2)$ | No privacy protection | $x_2$ will be exposed |
| Similarity-Conf Comp | $(x_1, x_2)$ | $P(y_2 = +1|x) \geq P(y_1 = +1|x)$ | Partial privacy protection | $x_2$ will be partially exposed |
| Similarity-Conf Diff | $(x_1, x_2, c)$ | $c = P(y_2 = +1|x) - P(y_1 = +1|x)$ | Partial privacy protection | $x_2$ will be partially exposed |
| USimUL (Ours) | $(x_1, x_2, x_3)$ | $y_1, y_2$ is i.i.d | **Full privacy protection** | $x_2$ **and** $x_3$ **are protected** |

Table 14: The statistics of the experimental datasets, including benchmark datasets, real-world weakly supervised learning (WSL) datasets, and real-world privacy-sensitive (Privacy) datatsets. Here, 5-C and 2-F denotes the neural networks with 5 convolutional layers and 2 fully-connected layers.

| Type | Dataset | #Training | #Testing | #Dim | Model |
|---|---|---|---|---|---|
| Benchmark | MNIST | 60K | 10K | 784 | MLP |
| | Fashion | 60K | 10K | 784 | MLP |
| | Kuzushi | 60K | 10K | 784 | MLP |
| | CIFAR-10 | 50K | 10K | 3072 | ResNet-34 |
| | SVHN | 73257 | 26032 | 3072 | ResNet-34 |
| Real-world WSL | Pendigits | 8793 | 2199 | 16 | MLP |
| | Lost | 418 | 104 | 50 | MLP |
| | MSRCv2 | 463 | 128 | 48 | MLP |
| | BirdSong | 4998 | 4994 | 38 | MLP |
| | Yahoo!News | 7813 | 1955 | 163 | MLP |
| Real-world Privacy | PDMD | 646 | 158 | 12288 | 5-C and 2-F |
| | PDSD | 740 | 185 | 12288 | 5-C and 2-F |
| | DDSM | 4080 | 1020 | 12288 | 5-C and 2-F |

sification tasks by using techniques such as ECOC (Dietterich & Bakiri, 1995), which transform traditional multi-class tasks into binary classification problems. In future work, we will attempt to extend the current approach to multi-class classification tasks.

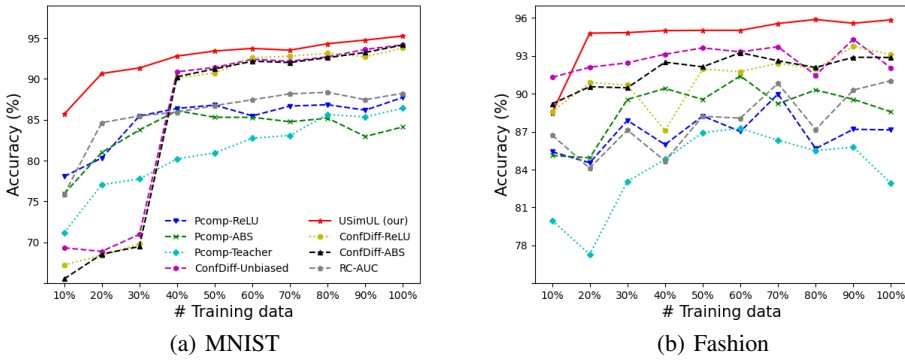

(a) MNIST

(b) Fashion

Figure 8: The classification accuracy of various methods when the amount of training data increases (under $\pi_+ = 0.6$).

