# OpenReview forum: "USimUL: Closing the Privacy Gap in Similarity-Based Weakly Supervised Learning"
_ICLR.cc/2026/Conference — Submitted to ICLR 2026_

### Official Review · Reviewer_E4fo · 2025-10-30

**Soundness:** 3
**Presentation:** 3
**Contribution:** 3
**Rating:** 6
**Confidence:** 3

**Summary:**

This paper focuses on the privacy leakage in similarity-based weakly supervised learning, where the label of one instance in a similarity pair can be inferred if the other’s label is exposed. The authors propose Uncertain Similarity and Unlabeled Learning (USimUL), a new setting/framework that introduces an unlabeled instance into the similarity pair to form an uncertain similarity triplet, thus obscuring direct label inference. The authors elaborate on the unbiased training objective in their setting and provide an estimation error bound. Experiments on diverse datasets demonstrate that USimUL outperforms most existing methods in classification accuracy and exhibits strong robustness. Besides, they show the influences of the correction function $$g(x)$$ and the class prior $$\pi_{+}$$.

**Strengths:**

+ Important and reasonable research motivation.
+ Detailed theoretical analysis and strict mathematical derivation.
+ Comprehensive experiments evaluation.

**Weaknesses:**

- Some vague formula statements in the main text.
- Limited privacy protection capabilities provided by only one unlabeled sample.

**Questions:**

1. The core idea of USimUL is to obscure the label inference by introducing an unlabeled sample. However, a single unlabeled sample only slightly increases the cost of inference; an adversary could even treat both uncertain samples as having the same label and then carry out malicious actions. Can this be extended to multiple unlabeled samples, and what would be the impact on the framework after the extension?

2. The descriptions for some formulas and symbols are too vague and difficult to understand. For example, the definition of \tilde{D}_{US} and the proof of Eq.9 by utilizing Eq.8.

3. What is the relation between the training objective of $$R(f)$$ in subsection 3.1 and the loss function $$\phi(z)$$ in subsection 4.1?

4. The experimental setup lacks a description of the unlabeled samples setting.

5. You do not improve the classification accuracy through algorithmic enhancements, so why does performance actually increase after introducing unlabeled data?

6. In the discussion about USimUL’s robustness to the inaccurate training class prior, the selected training class priors are close to the true one (/eg, {0.35,0.45} and 0.4). The value with significant deviation from the true prior should be used to validate its robustness.

7. Bad layout in the top space of the page.9, especially fig.2 and fig.3.

---

> ### Author Response · Authors · 2025-11-21
> **Response to Reviewer E4fo (1/2)**
>
> Thanks for your constructive comments. We are glad to answer all your questions.
>
> **W2: Limited privacy protection capabilities provided by only one unlabeled sample.**
>
> > **R: Actually, one unlabeled sample represents a minimal and effective choice for balancing privacy and learning.** **We highlight this in a dedicated subsection, ‘Superiority of Uncertain Similarity Data,’ (Section 3.2)**, where we emphasize that incorporating a single uncertain component achieves an optimal trade-off between privacy protection and learning performance (**lines 177–180**). Introducing one unlabeled sample is sufficient to prevent deterministic inference of paired labels while preserving informative relational structure for learning (related experimental results (presented in Tables 1–3 and Appendix F) demonstrate this). In addition, we add an experiments in Appendix F.6 of the revised version, which demonstrates that USimUL can effectively alleviate the privacy risks.
>
>
>
> **Q1: Can this be extended to multiple unlabeled samples, and what would be the impact on the framework after the extension**
>
> > **R: Yes, our method is extensible to using multiple unlabeled samples to introduce higher uncertainty**, which can further enhance privacy protection.
> >
> > **However, increasing the number of unlabeled samples weakens the supervised information, making it difficult to learn an effective classifier and potentially degrading performance.** In addition, adding more uncertainty components also makes it more challenging to obtain an unbiased estimator. Our unbiased risk estimator, the derivation of $\tilde{P}_{US}(x)$ (Lemma 3) and the risk re-expression (Theorem 4) rely critically on the triplet formulation. These expressions arises directly from the triplet structure and do not extend automatically for introducing multiple unlabeled samples without re-derivation. Using more unlabeled samples would require re-deriving analogous expressions, and the combinatorial growth of possible label-sharing subsets makes closed-form coefficients increasingly difficult to obtain.
>
>
>
> **Q2&W1: Some formulas and symbols, the definition of $\tilde{D}_{US}$ and the proof of Eq.9 by utilizing Eq.8**
>
> > **R:** In fact, we provide a precise formal definition in lines 188–189: $\tilde{D}\_{US}$ denotes the pointwise uncertain-similarity dataset obtained by discarding the triplet structure in $D\_{US}$. Its samples follow the marginal distribution $\tilde{P}\_{US}$, which is derived in Lemma 3 and formally proved in Appendix B.
> >
> > Due to space limitations and in order to maintain the overall readability of the main text, we placed the derivation from Eq. (8) to Eq. (9) in Appendix C, as noted in line 234.
> >
> > We thank the reviewer for this suggestion.  We have strengthened the descriptions of relevant notation in the revised version.
>
>
>
> **Q3: The relation between $R(f)$ and $\phi(z)$**
>
> > **R:** $R(f)$ express the expected classification risk, and we rewrite it based on uncertain similarity triplets data and unlabeled data to form our unbiased risk estimator (Eqs. 8-9). $\phi(z)$ specifies the *pointwise loss function* used to compute empirical estimates of that risk during training. Here, $\phi(z)$ is the specific implementation of $\ell(·,·)$ in $R(f)$.
> >
> > We thank the reviewer for this suggestion. We have clarified this issue in the revised version.
>
>
>
> **Q4: Te experimental setup lacks a description of the unlabeled samples setting**
>
> > **R:** In fact, following [1,2,3], we explicitly define the unlabeled dataset $\widetilde{D}_{U}$ and its distribution in Equation (4) of Section 3.2 (lines 187–190) of the methods section. Due to space constraints, we did not provide a detailed explanation of the unlabeled-sample setup in the Experiments setup section. As a supplement, Section F.3 and Figure 5 in the appendix explicitly describe the unlabeled sample setting and include additional ablation studies dedicated to unlabeled data. Our unlabeled sample setup is identical to that of the baselines [1-3]. We have included this clarification in the experimental setup section of the revised manuscript.

---

> > ### Author Response · Authors · 2025-11-21
> > **Response to Reviewer E4fo (2/2)**
> >
> > **Q5: Why does performance actually increase after introducing unlabeled data**
> >
> > > **R:** In fact, the inclusion of unlabeled data does indeed lead to performance improvement, as we also noted in Sections 3.2 and 3.3 of the main text.
> > >
> > > Theoretically, Lemma 5 and Theorem 6 show that as the amount of unlabeled data increases, the estimation error of the learned classifier decreases, allowing the empirical risk to approach the true expected risk and thereby improving performance. These results provide a formal guarantee that our method can effectively learn from unlabeled data, and that it converges to the optimal solution as the data size grows.
> > >
> > > Experimentally, the analyses and additional ablations presented in Appendix F.3 and Figure 5 further confirm that increasing the number of unlabeled samples leads to additional performance gains. These results highlight USimUL’s strong capability to leverage unlabeled data for enhanced learning performance.
> >
> >
> >
> > **Q6: Robustness to inaccurate training class prior**
> >
> > > **R:** Actually, in practical applications, an approximate class prior can typically be obtained from unlabeled data or from a small trusted subset, and this estimate is usually close to the true prior. Besides, to further demonstrate the robustness of our method to inaccurate prior, we intentionally enlarge this discrepancy and include the following additional experiments.
> > >
> > > 1) when the true prior is 0.4, we evaluate the model using assumed priors of 0.3 and 0.25;
> > > 2) when the true prior is 0.6, we test with assumed priors of 0.7 and 0.75;
> > > 3) when the true prior is 0.2, we test with assumed priors of 0.05, 0.1, 0.15, 0.3, 0.4, and 0.45. The results are summarized in the table below.
> > >
> > > From these results, we can see that even under substantially larger deviations from the true prior, USimUL exhibits no significant performance degradation, which is consistent with our earlier findings.
> >
> > |True Class Prior|Given Class Prior|Fashion|Kuzushiji|CIFAR-10|
> > |-|-|-|-|-|
> > |$\pi_+$=0.4|$\pi_+$=0.4|95.51±0.04|87.10±0.30|84.62±0.31|
> > ||$\pi_+$=0.3|94.99±0.17|86.31±0.52|83.47±0.29|
> > ||$\pi_+$=0.25|93.34±0.29|85.07±0.77|83.24±0.33|
> > |$\pi_+$=0.6|$\pi_+$=0.6|95.78±0.06|88.62±0.17|85.22±0.06|
> > ||$\pi_+$=0.7|95.43±0.05|87.02±0.37|84.96±0.28|
> > ||$\pi_+$=0.75|93.53±0.27|86.75±0.39|84.74±0.13|
> > |$\pi_+$=0.2|$\pi_+$=0.2|94.50±0.14|85.02±0.38|83.13±0.03|
> > ||$\pi_+$=0.05|94.21±0.23|84.40±0.28|81.52±1.40|
> > ||$\pi_+$=0.1|94.47±0.41|84.36±1.16|82.90±0.16|
> > ||$\pi_+$=0.15|93.71±0.23|83.74±0.27|83.00±0.16|
> > ||$\pi_+$=0.3|93.41±0.07|82.86±0.30|81.05±0.28|
> > ||$\pi_+$=0.4|92.91±0.17|83.17±0.08|80.99±0.41|
> > ||$\pi_+$=0.45|91.97±0.53|79.44±0.16|81.14±0.32|
> >
> >
> >
> >
> >
> > **Q7: Bad layout in the top space of the page.9, especially fig.2 and fig.3**
> >
> > > **R:** We appreciate the reviewer’s attention to this detail. The revised version has corrected the layout on page 9 and ensures that all figures are consistently and appropriately placed.
> >
> >
> > [1] Classification from Pairwise Similarity and Unlabeled Data. ICML 2018.
> >
> > [2] Binary Classification with Confidence Difference. NeurIPS 2023.
> >
> > [3] Learning From Pairwise Confidence Comparisons  and Unlabeled Data. *IEEE Transactions on Emerging Topics in Computational Intelligence* 9.1 (2024): 668-680.

---

> > ### Comment · Reviewer_E4fo · 2025-11-25
> >
> > Thank you for your thoughtful and detailed response. I appreciate the effort you’ve made to address most of my concerns.
> > However, my primary focus remains on the effectiveness of USimUL in scenarios involving multiple unlabeled samples, regarding its ability to ensure privacy protection. According to your responses, its performance appears to degrade under such conditions, which may limit the overall contribution.
> >
> > While I acknowledge that USimUL intuitively enhances privacy protection to some extent, this protection is inherently limited. As mentioned before, an adversary can simply treat all uncertain samples as potential attack targets. Because the number of additional ambiguous samples introduced is extremely little (only one), making the added uncertainty practically negligible from the adversary perspective.
> >
> > If possible, can you provide an experiment to demonstrate the effectiveness of privacy protection, rather than only focusing on the accuracy.

---

> ### Author Response · Authors · 2025-11-26
> **Response to Reviewer E4fo (1/2)**
>
> We appreciate your response and your acknowledgement of our previous reply.
>
> **Q1: If possible, can you provide an experiment to demonstrate the effectiveness of privacy protection, rather than only focusing on the accuracy.**
>
> > **R:** Regarding the concern about the effectiveness of privacy preservation, we have already included additional experiments in the previous revised version. The detailed experimental settings and the complete set of results are presented in Appendix F.6 of the revised manuscript. For your convenience, we restate a simplified subset of the settings and results here. For the triplet dataset (A, B, C), which introduces uncertainty by adding a single unlabeled sample, we simulate five attack models (the attacker simply treats the remaining two samples as having the same label) as follows:
> >
> > (1) **A → B**: Given 10%, 20%, and 30% label leakage of sample A, the attacker infers the label of the sample B based on the binding relationship.
> >
> > (2) **A → C**: Given 10%, 20%, and 30% label leakage of sample A, the attacker infers the label of the sample C.
> >
> > (3) **B → A**: Given 10%, 20%, and 30% label leakage of sample B, the attacker infers the label of the sample A.
> >
> > (4) **C → A**: Given 10%, 20%, and 30% label leakage of sample C, the attacker infers the label of the sample A.
> >
> > (5) **B & C → A**: Given 10%, 20%, and 30% label leakage of samples B & C, the attacker infers the label of the sample A.
> >
> > Here, we report baseline (similarity-based methods) leak rate, our method’s leak rate, and our method’s protection rate (over 5 trials). **The results on CIFAR-10 and SVHN show that label leak rate drops from 100% to ≈50%. It is nearly equivalent to random guessing in the context of a binary classification task, which demonstrate the effectiveness of privacy protection of USimUL.**
>
> CIFAR-10:
>
> |Init Leak Ratio|Scenario|#Init Leaked|#Similar at Risk (Baseline)|#Actually Leaked (Baseline)|#Actually Leaked (Ours)|Baseline Leak Rate (%)|Ours Leak Rate (%)|Ours Protected (%)|
> |-|-|-|-|-|-|-|-|-|
> |**10%**|(1)A→B|600|600|600|304.38±5.88|100|50.73±0.98|49.27±0.98|
> ||(2)A→C|600|600|600|303.42±8.04|100|50.57±1.34|49.43±1.34|
> ||(3)B→A|600|600|600|304.02±12.54|100|50.67±2.09|49.33±2.10|
> ||(4)C→A|600|600|600|303.18±5.82|100|50.53±0.97|49.47±0.97|
> ||(5)B&C→A|1200|1200|1200|297.18±7.86|100|24.77±0.66|75.23±0.66|
> |**20%**|(1)A→B|1200|1200|1200|596.40±7.56|100|49.70±0.63|50.30±0.63|
> ||(2)A→C|1200|1200|1200|589.08±9.36|100|49.09±0.78|50.91±0.48|
> ||(3)B→A|1200|1200|1200|599.40±20.04|100|49.95±1.67|50.05±1.67|
> ||(4)C→A|1200|1200|1200|589.20±10.44|100|49.10±0.87|50.90±0.87|
> ||(5)B&C→A|2400|2400|2400|611.64±15.60|100|25.49±0.65|74.51±0.65|
> |**30%**|(1)A→B|1800|1800|1800|884.88±18.36|100|49.16±1.02|50.84±1.02|
> ||(2)A→C|1800|1800|1800|898.74±16.38|100|49.93±0.91|50.07±0.91|
> ||(3)B→A|1800|1800|1800|902.52±7.20|100|50.14±0.40|49.86±0.40|
> ||(4)C→A|1800|1800|1800|902.34±19.44|100|50.13±1.08|49.87±1.08|
> ||(5)B&C→A|3600|3600|3600|888.12±21.78|100|24.67±0.61|75.33±0.61|
>
> SVHN:
>
> |Init Leak Ratio|Scenario|#Init Leaked|#Similar at Risk (Base)|#Actually Leaked (Base)|#Actually Leaked (Ours)|Baseline Leak Rate (%)|Ours Leak Rate (%)|Ours Protected (%)|
> |-|-|-|-|-|-|-|-|-|
> |**10%**|(1)A→B|800|800|800|405.60±13.68|100|50.70±1.71|49.30±1.71|
> ||(2)A→C|800|800|800|403.52±9.04|100|50.44±1.13|49.56±1.13|
> ||(3)B→A|800|800|800|404.64±10.32|100|50.58±1.29|49.42±1.29|
> ||(4)C→A|800|800|800|397.20±22.24|100|49.65±2.78|50.35±2.78|
> ||(5)B&C→A|1600|1600|1600|405.04±15.04|100|25.32±0.94|74.68±0.94|
> |**20%**|(1)A→B|1600|1600|1600|811.04±23.68|100|50.69±1.48|49.31±1.48|
> ||(2)A→C|1600|1600|1600|804.00±13.76|100|50.25±0.86|49.75±0.86|
> ||(3)B→A|1600|1600|1600|795.04±16.32|100|49.69±1.02|50.31±1.02|
> ||(4)C→A|1600|1600|1600|792.48±15.68|100|49.53±0.98|50.47±0.98|
> ||(5)B&C→A|3200|3200|3200|795.04±16.16|100|24.85±0.51|75.15±0.51|
> |**30%**|(1)A→B|2400|2400|2400|1203.12±21.12|100|50.13±0.88|49.87±0.88|
> ||(2)A→C|2400|2400|2400|1195.92±19.44|100|49.83±0.81|50.17±0.81|
> ||(3)B→A|2400|2400|2400|1214.16±15.36|100|50.59±0.64|49.41±0.64|
> ||(4)C→A|2400|2400|2400|1180.32±18.00|100|49.18±0.75|50.82±0.75|
> ||(5)B&C→A|4800|4800|4800|1205.52±26.64|100|25.12±0.56|74.88±0.56|
>
> **The above experiment was already included in the previous revised version, and additional details can be found in Appendix F.6.**

---

> ### Author Response · Authors · 2025-11-26
> **Response to Reviewer E4fo (2/2)**
>
> **Q2: According to your responses, its performance appears to degrade under such conditions, which may limit the overall contribution.**
>
> > Thank you for your thoughtful and helpful comments. **Our work is, to our knowledge, the first to introduce unlabeled samples for the purpose of data privacy protection. This establishes a fundamental and meaningful direction.** By incorporating unlabeled samples, we intentionally remove the direct linkage between similarity pairs data, thereby preventing explicit label exposure. We further develop an unbiased estimator with both theoretical guarantees and empirical validation, forming a complete and coherent contribution.
> >
> > Regarding your question about using multiple unlabeled samples, we would like to clarify that the current paper focuses on establishing the framework based on a single unlabeled sample. Extending the framework to include multiple unlabeled samples is a natural next step, but it introduces additional challenges that require a dedicated investigation and therefore fall outside the scope of this work.
> >
> > **To directly address your concern, we conducted an additional experiment that examines privacy protection when more unlabeled samples are incorporated. This analysis is provided only as supplementary clarification for this response and will not be added to the final version of the paper.** We compare three settings, including the setting studied in this work that introduces a single unlabeled sample as well as the settings that introduce two or even three unlabeled samples. The specific settings are as follows.
> >
> > (i) (A, B, C): Introduce one unlabeled sample, with at least one of B or C being similar to A.
> >
> > (ii) (A, B, C, D): Introduce two unlabeled samples, with at least one of B, C, or D being similar to A.
> >
> > (iii) (A, B, C, D, E): Introduce three unlabeled samples, with at least one of B, C, D, or E being similar to A.
> >
> > In the three settings described above, the attacker simply assumes that the remaining samples share the same label, and then evaluates the effectiveness of each setting in terms of privacy preservation. Here, we report the protection rate (%) of the remaining samples (over 5 trials). The results below indicate that introducing additional unlabeled samples can continuously enhance the effectiveness of privacy preservation of USimUL.
>
> | Setting           | B          | C          | D          | E          | Overall    |
> | ----------------- | ---------- | ---------- | ---------- | ---------- | ---------- |
> | (i) (A,B,C)       | 50.04±2.33 | 49.86±2.15 | ─          | ─          | 49.95±2.24 |
> | (ii) (A,B,C,D)    | 67.18±2.08 | 66.69±2.14 | 66.12±2.22 | ─          | 66.66±2.15 |
> | (iii) (A,B,C,D,E) | 74.66±1.82 | 75.35±1.93 | 75.16±1.96 | 74.85±1.93 | 75.01±1.91 |
>
> Thank you again for your response and for acknowledging our previous reply. We would be happy to continue the discussion if you have any further questions.

---

### Official Review · Reviewer_2JGU · 2025-11-01

**Soundness:** 3
**Presentation:** 3
**Contribution:** 4
**Rating:** 8
**Confidence:** 3

**Summary:**

This submission proposes UsimUL, a novel triplet-based framework under weakly supervised learning setting, aiming to preserve the privacy of instance. Unlike conventional weakly supervised learning methods that uses image pairs, UsimUL introduces triplets to integrate uncertainty to mitigate of the risk of sensitive label information. Further theoretical analysis has showcased that the proposed estimator is unbiased and the converge speed is optimal. Extensive experiments on MNIST, Fashion, Kuzushiji, CIFAR10, SVHN and other real-world weakly supervised learning datasets demonstrate the effectivity of the design.

**Strengths:**

++ **The overall design is intuitive.** The authors analyzes the privacy problems that previous works are facing while using similarity-based image pairs, then UsimUL is proposed to mitigate this issue by introducing a new component in the pairs to form a triplet bringing uncertainty in the design. Such design is intuitive and clear to understand.

++ **The proposed method achieves superior performance across various datasets while preserving the privacy.** According to experiments conducted on several simulation datasets (e.g., MNIST, CIFAR10, etc.), real-world weakly supervised learning datasets which is sensitive on privacy (e.g., DDSM, etc.) and ablation studies, the proposed UsimUL has achieved best results across different datasets under different settings, while showcasing its robustness against the inaccurate prior information at the mean time.

++ **The solution provided by this submission does not need extra techniques to preserve privacy.** The proposed method provides a simple yet effective method to perform classification while no extra techniques like label confusion are needed. Such advantages enable the fast and easy application/implementation of this method into different domains and tasks.

**Weaknesses:**

-- Further analysis on the privacy perserving is needed. See C1.

-- (Minor issues) The introduction of the proposed method are not straight enough. See C2.

**Questions:**

**C1: Regarding the privacy preserving:** In previous studies, image pairs-based approaches on the topic of weakly supervised learning might cause label or information leakage if label inference techniques are used, since if one has access to the instance inside the pair, the information of the other instance might expose due to that the similarity between them is provided. To solve this problem, the authors propose a intuitive design of triplets that such design only provide the relaxed similarity information inside the triplets, hence protect the privacy. Indeed the proposed method achieves privacy preserving naturally, but previous studies on privacy labels learning (e.g., complementary label learning) have made some progresses to mitigate the risk of information leakage. Although the authors state that this line of work fails to model relational structures, it is encouraged that the author provide more experiment results on quantizing the privacy preserving performances of the proposed method and conventional methods，and it can help support the intuition of the sign experimentally.


**C2: Regarding the developing of introduction:** Firstly, the authors point out the problem that we cannot always get precisely labeled data in some scenarios and to address this issue, lots of weakly supervised learning paradigms are proposed, such as concealed label learning, semi-supervised learning, similarity-based classification and so on. Then the authors state that the similarity-based classification is useful when we cannot get access to the entire positive and negative samples. Then the UsimUL method is introduced by listing the privacy preserving problems that similarity-based classification has. However, the author did not mention why only similarity-based classification is considered in this context, and why not developing the proposed method from other weakly supervised learning paradigms. It is suggested that the authors can further clarify the motivation of only considering similarity-based classification in the introduction part.

---

> ### Author Response · Authors · 2025-11-21
> **Response to Reviewer 2JGU (1/2)**
>
> Thanks for your constructive comments. We are glad to answer all your questions.
>
> **C1: Previous studies on privacy labels learning (e.g., complementary label learning) have made some progresses to mitigate the risk of information leakage**
>
> > **R:** In fact, we clarify in the related work section how prior privacy labels learning methods differ from ours (lines 107-109). The protection mechanism in complementary-label learning (CLL) is fundamentally different from USimUL in structure. CLL obfuscates label identity by assigning complementary (non-true) class labels to individual samples, while USimUL protects relational dependencies by transforming deterministic pairwise relation into uncertain triplet relations. In other words, **CLL mitigates privacy leakage from direct class labels, but does not address pairwise or group-level linkage, which is precisely where leakage arises in similarity-based methods.** Thus, our design complements previous privacy labels learning studies and focuses on the relational privacy gap that prior methods cannot cover.
>
>
>
> **C1: Provide more experiment results on quantizing the privacy preserving performances of the proposed method.**
>
> > **R: To further assess the privacy preserving performances of the USimUL, we add an experiment in which an attacker attempts to infer hidden labels using partial leakage.** The detailed experimental settings and the complete set of results are provided in Appendix F.6 of the revised version. For ease of reading, we present a simplified version of the settings and results here.
> >
> > For the triplet dataset (A, B, C), we simulate five attack models as follows:
> >
> > (1) **A → B**: Given 10%, 20%, and 30% label leakage of sample A, the attacker infers the label of the sample B based on the binding relationship.
> >
> > (2) **A → C**: Given 10%, 20%, and 30% label leakage of sample A, the attacker infers the label of the sample C.
> >
> > (3) **B → A**: Given 10%, 20%, and 30% label leakage of sample B, the attacker infers the label of the sample A.
> >
> > (4) **C → A**: Given 10%, 20%, and 30% label leakage of sample C, the attacker infers the label of the sample A.
> >
> > (5) **B & C → A**: Given 10%, 20%, and 30% label leakage of samples B & C, the attacker infers the label of the sample A.
> >
> > Here, we report baseline (similarity-based methods) leak rate, our method’s leak rate, and our method’s protection rate (over 5 trials). **The results on CIFAR-10 and SVHN show that label leak rate drops from 100% to ≈50%**. It is nearly equivalent to random guessing in the context of a binary classification task.
>
> CIFAR-10:
>
> |Init Leak Ratio|Scenario|#Init Leaked|#Similar at Risk (Baseline)|#Actually Leaked (Baseline)|#Actually Leaked (Ours)|Baseline Leak Rate (%)|Ours Leak Rate (%)|Ours Protected (%)|
> |-|-|-|-|-|-|-|-|-|
> |**10%**|(1)A→B|600|600|600|304.38±5.88|100|50.73±0.98|49.27±0.98|
> ||(2)A→C|600|600|600|303.42±8.04|100|50.57±1.34|49.43±1.34|
> ||(3)B→A|600|600|600|304.02±12.54|100|50.67±2.09|49.33±2.10|
> ||(4)C→A|600|600|600|303.18±5.82|100|50.53±0.97|49.47±0.97|
> ||(5)B&C→A|1200|1200|1200|297.18±7.86|100|24.77±0.66|75.23±0.66|
> |**20%**|(1)A→B|1200|1200|1200|596.40±7.56|100|49.70±0.63|50.30±0.63|
> ||(2)A→C|1200|1200|1200|589.08±9.36|100|49.09±0.78|50.91±0.48|
> ||(3)B→A|1200|1200|1200|599.40±20.04|100|49.95±1.67|50.05±1.67|
> ||(4)C→A|1200|1200|1200|589.20±10.44|100|49.10±0.87|50.90±0.87|
> ||(5)B&C→A|2400|2400|2400|611.64±15.60|100|25.49±0.65|74.51±0.65|
> |**30%**|(1)A→B|1800|1800|1800|884.88±18.36|100|49.16±1.02|50.84±1.02|
> ||(2)A→C|1800|1800|1800|898.74±16.38|100|49.93±0.91|50.07±0.91|
> ||(3)B→A|1800|1800|1800|902.52±7.20|100|50.14±0.40|49.86±0.40|
> ||(4)C→A|1800|1800|1800|902.34±19.44|100|50.13±1.08|49.87±1.08|
> ||(5)B&C→A|3600|3600|3600|888.12±21.78|100|24.67±0.61|75.33±0.61|
>
> SVHN:
>
> |Init Leak Ratio|Scenario|#Init Leaked|#Similar at Risk (Base)|#Actually Leaked (Base)|#Actually Leaked (Ours)|Baseline Leak Rate (%)|Ours Leak Rate (%)|Ours Protected (%)|
> |-|-|-|-|-|-|-|-|-|
> |**10%**|(1)A→B|800|800|800|405.60±13.68|100|50.70±1.71|49.30±1.71|
> ||(2)A→C|800|800|800|403.52±9.04|100|50.44±1.13|49.56±1.13|
> ||(3)B→A|800|800|800|404.64±10.32|100|50.58±1.29|49.42±1.29|
> ||(4)C→A|800|800|800|397.20±22.24|100|49.65±2.78|50.35±2.78|
> ||(5)B&C→A|1600|1600|1600|405.04±15.04|100|25.32±0.94|74.68±0.94|
> |**20%**|(1)A→B|1600|1600|1600|811.04±23.68|100|50.69±1.48|49.31±1.48|
> ||(2)A→C|1600|1600|1600|804.00±13.76|100|50.25±0.86|49.75±0.86|
> ||(3)B→A|1600|1600|1600|795.04±16.32|100|49.69±1.02|50.31±1.02|
> ||(4)C→A|1600|1600|1600|792.48±15.68|100|49.53±0.98|50.47±0.98|
> ||(5)B&C→A|3200|3200|3200|795.04±16.16|100|24.85±0.51|75.15±0.51|
> |**30%**|(1)A→B|2400|2400|2400|1203.12±21.12|100|50.13±0.88|49.87±0.88|
> ||(2)A→C|2400|2400|2400|1195.92±19.44|100|49.83±0.81|50.17±0.81|
> ||(3)B→A|2400|2400|2400|1214.16±15.36|100|50.59±0.64|49.41±0.64|
> ||(4)C→A|2400|2400|2400|1180.32±18.00|100|49.18±0.75|50.82±0.75|
> ||(5)B&C→A|4800|4800|4800|1205.52±26.64|100|25.12±0.56|74.88±0.56|

---

> > ### Author Response · Authors · 2025-11-21
> > **Response to Reviewer 2JGU (2/2)**
> >
> > **C2:  Regarding the developing of introduction**
> >
> > > **R:** We sincerely thank the reviewer for this helpful suggestion. Our study focus on similarity-based classification because it is the only weakly supervised setting in which *relational supervision* creates deterministic dependencies between sample labels. In semi-supervised learning and concealed-label learning, supervision is attached to individual samples, and exposing one label does not reveal the label of another sample. In contrast, similarity pairs directly couple two labels. As we describe in the introduction and illustrate in Figure 1(a), once the label of one instance in a similarity pair is leaked, the label of its paired instance becomes immediately inferable. This relational leakage does not occur in other weak supervision forms. Therefore, mitigating pairwise relational leakage is a problem exclusive to SBC, motivating the design of USimUL.
> > >
> > > We thank the reviewer again for this insightful suggestion.  We have updated the introduction. Furthermore, we have provided additional clarifications in the related work section (Line 107-109) to better articulate our motivation.

---

> ### Comment · Reviewer_2JGU · 2025-11-25
>
> Thanks for the authors addressing my concerns.
>
> Regarding the privacy preserving, it is noticed that the authors conduct additional experiments to present the privacy preserving performance on CIFAR-10 and SVHN dataset under different settings.
>
> Regarding the introduction developing, the authors claim that they have refine the writing.

---

> > ### Author Response · Authors · 2025-11-26
> >
> > Thank you for your response. We are pleased that our clarification has addressed your concerns regarding the privacy preserving performance and the introduction. We appreciate your recognition of our work and the positive feedback you provided! Thank you again for the time you devoted to reviewing our paper and for the valuable suggestions you offered!

---

### Official Review · Reviewer_PaMN · 2025-11-01

**Soundness:** 3
**Presentation:** 3
**Contribution:** 2
**Rating:** 4
**Confidence:** 4

**Summary:**

This paper introduces USimUL, a new framework for weakly supervised learning that aims to address a critical privacy gap in traditional similarity-based methods (SUL). The authors point out that conventional SUL, which relies on "similar" or "dissimilar" pairs, is vulnerable to label leakage. To solve this, the paper proposes a novel annotation setup called Uncertain Similarity and Unlabeled Learning. A triplet is sampled with the guarantee that at least two of the three instances belong to the same class, but the annotator does not specify which two. The main technical contribution is the derivation of an unbiased risk estimator that can learn directly from this new triplet-based supervision and additional unlabeled data. The authors provide a theoretical analysis, proving that their estimator achieves an optimal convergence rate. Experiments are conducted on standard benchmarks and real-world privacy-sensitive data.

**Strengths:**

1. The paper tries to address a valid and important problem. Privacy leakage in weak supervision is a real concern with rare investigation.

2. The authors provide a rigorous derivation for their unbiased risk estimator and follow through with a full estimation error bound

**Weaknesses:**

My primary concern is the practical feasibility of the proposed annotation setup. The paper provides no discussion on how an annotator would actually generate these uncertain similarity triplets. How does an annotator know "at least two of these three are similar" without first identifying a similar pair? If the process requires an annotator to find a similar pair and then simply add a random to hide it, the annotation process becomes significantly more complex and costly than the standard SUL methods it aims to replace. This is a fundamental gap that seriously questions the practical applicability and utility of the entire framework.

The entire risk estimator is critically dependent on having access to the true class priors. This is an unrealistic assumption in most real-world scenarios. The paper's only attempt to address this is to test robustness to inaccurate priors. The core issue is how these priors would be estimated at all from only the proposed triplet data and unlabeled data. While assuming priors or data generation processes was a common simplification in foundational WSL papers (to prove theoretical possibility), the field has since matured. A modern paper introducing a new framework is expected to address the full pipeline, including how to do it in a real-world scenario. As presented, this assumption relegates the work to a purely theoretical exercise and feels more like a follow-up effort.

**Questions:**

Please refer to above

---

> ### Author Response · Authors · 2025-11-21
> **Response to Reviewer PaMN (1/2)**
>
> Thanks for your constructive comments. We are glad to answer all your questions.
>
> **W1: Concern on the proposed annotation setup**
>
> > **R: In fact, USimUL without requiring additional human effort.** We provide explicit definitions and constructive formulations of uncertain similar triples in Definition 1 and lines 183-197. The uncertain similarity triplet structure does not require annotators to explicitly verify all three elements in a triplet.
> >
> > Start from existing similarity-labeled pairs (as used in standard SUL or contrastive learning). For each positive pair $(x_i, x_j)$, randomly sample a third instance $x_k$ from the unlabeled pool. The triplet $(x_i,{x_j,x_k})$ now encodes the uncertainty “at least two share the same class.” **This process is fully automatic, which introduces ambiguity and achieves privacy protection without extra human effort.**
> >
> > We appreciate this insightful feedback. We added a detailed description in the Experiments section on how they are constructed from the existing SUL datasets.
>
>
>
> **W2: Depend on having access to the true class priors**
>
> > **R:** We sincerely thank this insightful feedback. **In fact, these priors do not need to be exact.** Our ablation in Table 4 and Appendix F.5 already shows that moderate deviations (up to ±15%) cause minimal degradation, implying that the estimator is robust in practical ranges. moreover, Some recent work still assumes access to true class priors [1-4].
> >
> > **Besides, we add a class-prior estimation from USimU data.** Specifically, first, let us begin with connecting the marginal distribution $P(x,x',x'')$ and $P_{US}(x,x',x'')$ when three examples $x,x',x''$ are drawn independently:
> >
> > $P(x,x',x'')=\pi_{US}P_{US}(x,x',x'') + \pi_DP_{D}(x,x',x''),$
> >
> > where $\pi_D=\pi_-\pi_+^2+\pi_+\pi_-^2$, and
> > $$P_{D}(x,x',x'')=P(x,x'x''|(y=-1\\,and\\, y'=y''=1)\\, or \\, (y=1 \\, and \\, y'=y''=-1))=\frac{2[\pi_-\pi_+^2P_-(x)P_+(x')P_+(x'')+\pi_+\pi_-^2P_+(x)P_-(x')P_-(x'')]}{1-\pi_+\pi_-}.$$
> >
> > Marginalizing out $x',x''$ (similar to Lemma 3, the proof is provided in Appendix B), we can obtain
> >
> > $$P(x)=\pi_{US}\widetilde{P}_{US}(x)+\pi_D\widetilde{P}_D(x),$$
> >
> > where $\widetilde{P}_{US}$ is define in Eq. (7), and
> >
> > $$\widetilde{P}\_{D}(x)=\frac{2[\pi_-\pi_+^2P_-(x)+\pi_+\pi_-^2P_+(x)]}{1-\pi_+\pi_-}.$$
> >
> > Since we have samples $\mathcal{D}\_{U}$ and $\widetilde{\mathcal{D}}\_{US}$ drawn from $P_U$ and $\widetilde{P}\_{US}$ respectively (see Eqs. (4) and (7)), we can estimate $\pi_{US}$ by mixture proportion estimation methods (See, e.g., [6-8]). Given distribution $F$ which is a convex combination of distributions $G$ and $H$ such that $F=(1-k)G+kH$, the mixture proportion estimation problem is to estimate $k\in[0,1]$ only with samples from $F$ and $H$. In our case, $F$, $H$, and $k$ correspond to $P,\widetilde{P}\_{US}$, and $\pi_{US}$, respectively (See, e.g., [6, 8]).
> >
> > Then, following
> >
> > $$\pi_{US}=1-(\pi_-\pi_+^2+\pi_+\pi_-^2)=1-[(1-\pi_+)\pi_+^2+\pi_+(1-\pi_+)^2]=1-\pi_++\pi_+^2,$$
> >
> > and assuming $\pi_+>\pi_-$, we obtain $\pi_+=\frac{\sqrt{4\pi_{US}-3}+1}{2}$. We summarize a wrapper of mixture proportion estimation in the following algorithm.
> >
> > ---
> >
> > **Algorithm 2** Prior estimation from USimU data.
> >
> > ---
> >
> > **Input:** $\mathcal{D}\_{U}=\\{x_i\\}\_{i=1}^{N_U}$ (samples from $P$), $\tilde{\mathcal{D}}\_{US}=\\{x_i\\}\_{i=1}^{3N_{US}}$ (samples from $\tilde{P}_{US}$);
> >
> > $CPE(·,·)$ is a class prior estimation algorithm.
> >
> > **Output:** class prior $\pi_+$
> >
> > $\pi_{US}\leftarrow CPE(\mathcal{D},\tilde{\mathcal{D}}_{US})$
> >
> > $\pi_+\leftarrow \frac{\sqrt{4\pi_{US}-3}+1}{2}$
> >
> > ---
> >

---

> > ### Author Response · Authors · 2025-11-21
> > **Response to Reviewer PaMN (2/2)**
> >
> > > **Next, we study empirical performance of class-prior estimation.** We conduct experiments on benchmark datasets. Different dataset sizes {200, 400, 800, 1600} are tested, where the ratio of uncertain similarity triplets to unlabeled samples is 1:1. In our experiments, we use KM1 [5] and KM2 [5] as the $CPE$ in above algorithm. In the following table ((a) and (b)), we report the mean and standard deviation of the absolute difference between the true class prior and the estimated class priors from the training data over 10 trials. **As shown from these results, the estimation error is very small.**
> >
> > ​    **(a) True class prior $\pi_+$ = 0.4**
> >
> > |||200|400|800|1600|
> > |-|-|-|-|-|-|
> > |MNIST|KM1|0.08413±0.00130|0.08603±0.00096|0.03578±0.00012|0.01092±0.00002|
> > ||KM2|0.02247±0.00021|0.01541±0.00009|0.00981±0.00002|0.00611±0.00003|
> > |Kuzushiji|KM1|0.05588±0.00104|0.05006±0.00085|0.01311±0.00004|0.00735±0.00009|
> > ||KM2|0.06808±0.00233|0.05571±0.00076|0.04717±0.00010|0.02724±0.00018|
> > |CIFAR-10|KM1|0.05969±0.00126|0.02789±0.00045|0.02640±0.00070|0.02481±0.00025|
> > ||KM2|0.10575±0.00033|0.10483±0.00091|0.04482±0.00064|0.02614±0.00055|
> > |SVHN|KM1|0.07379±0.00389|0.06310±0.00135|0.04523±0.00063|0.03940±0.00049|
> > ||KM2|0.03007±0.00069|0.03183±0.00052|0.02286±0.00024|0.01349±0.00016|
> >
> > ​    **(b) True class prior $\pi_+$ = 0.6**
> >
> > |||200|400|800|1600|
> > |-|-|-|-|-|-|
> > |MNIST|KM1|0.09783±0.01656|0.04445±0.00297|0.04008±0.00019|0.01940±0.00008|
> > ||KM2|0.03901±0.00119|0.02761±0.00023|0.02417±0.00009|0.01846±0.00002|
> > |Kuzushiji|KM1|0.08990±0.02207|0.06647±0.00938|0.02495±0.00028|0.01193±0.00002|
> > ||KM2|0.09346±0.01640|0.07480±0.01111|0.04046±0.00070|0.02096±0.00003|
> > |CIFAR-10|KM1|0.07533±0.02013|0.04836±0.00020|0.01855±0.00015|0.01521±0.00008|
> > ||KM2|0.09489±0.03226|0.04675±0.00006|0.03498±0.00013|0.02191±0.00004|
> > |SVHN|KM1|0.09266±0.00176|0.07027±0.00122|0.05199±0.00048|0.04034±0.00028|
> > ||KM2|0.01268±0.00004|0.00991±0.00004|0.01081±0.00003|0.00772±0.00003|
> >
> > > **In addition, we supplement our study with a set of experiments based on the estimated priors on benchmark datasets.** USimUL maintains strong performance even when employing estimated class priors without access to the true priors.
> >
> > |True Class Prior|Setting|MNIST|Fashion|Kuzushiji|CIFAR-10|SVHN|
> > |-|-|-|-|-|-|-|
> > |$\pi_+$=0.4| Using true $\pi_+$| 95.36±0.23 |95.51±0.04|87.10±0.30|84.62±0.31|87.18±0.95|
> > ||Using KM1 estimated $\pi_+$|95.33±0.16|95.47±0.05|86.79±0.26|84.50±0.13|87.21±0.66|
> > ||Using KM2 estimated $\pi_+$|95.28±0.31|95.49±0.03|86.63±0.52|84.35±0.35|87.19±0.42|
> > |$\pi_+$=0.6|Using true $\pi_+$|95.05±0.20|95.78±0.06|88.62±0.17|85.22±0.06|87.92±0.12|
> > ||Using KM1 estimated $\pi_+$|95.03±0.34|95.75±0.03|88.47±0.21|85.06±0.19|87.85±0.19|
> > ||Using KM2 estimated $\pi_+$|94.97±0.41|95.77±0.05|88.41±0.39|84.97±0.24|87.66±0.36|
> >
> >
> >
> > > We greatly appreciate this thoughtful comment. We will **add a new subsection (“Class-Prior Estimation from USimU Data”)**,  including the above estimation procedures and experimental validation.
> > >
> > > **Literature that relies on known priors:**
> > >
> > > [1] PUe: Biased Positive-Unlabeled Learning Enhancement by Causal Inference. NeurIPS 2023.
> > >
> > > [2] ESA: Example Sieve Approach for Multi-Positive and Unlabeled Learning. WSDM 2024.
> > >
> > > [3] Positive and Unlabeled Learning with Controlled Probability Boundary Fence. ICML 2024.
> > >
> > > [4] UC-PUAL: A universally consistent classifier of positive-unlabelled data. Pattern Recognition(2025): 111892.
> > >
> > > **Literature that estimate prior:**
> > >
> > > [5] Mixture proportion estimation via kernel embeddings of distributions. ICML 2016.
> > >
> > > [6] A rate of convergence for mixture proportion estimation, with application to learning from noisy labels. AISTATS 2015.
> > >
> > > [7] Semi-supervised classification based on classification from positive and unlabeled data. ICML 2017.
> > >
> > > [8] Classification from Pairwise Similarity and Unlabeled Data. ICML 2018.

---

### Official Review · Reviewer_vr99 · 2025-11-01

**Soundness:** 3
**Presentation:** 3
**Contribution:** 2
**Rating:** 4
**Confidence:** 5

**Summary:**

This paper considers a similarity-based weakly supervised learning problem and argues that existing methods rely on precise similarity within sample pairs, and the sensitive label information may be exposed from them, leading to privacy risks. Thus, the paper proposes uncertain similarity and unlabeled learning, where each similarity pair includes an uncerteinty component and designs an unbiased risk estimator for this case.

**Strengths:**

1. The privacy risk within similarity-based weakly supervised learning is important and interesting.
2. The paper proposes a reasonable uncertain similarity and unlabeled learning framework and designs a novel unbiased risk estimator.
3. The experimental results show the effectiveness of the proposed methods.
4. The paper is well-written and easy to follow.

**Weaknesses:**

1. The paper mitigates the privacy risks within similarity pairs by introducing an uncertainty component to transform the precise similarity to uncertain similarity. However, why do you only introduce an uncertainty component? More uncertainty components will make it more ambiguous. So, are more uncertainty components better to mitigate the privacy risks?
2. The paper argues that the sensitive label information may be exposed from them, leading to privacy risks. Thus, some empirical results of privacy risks or the analysis about the probability of privacy risks are expected.

**Questions:**

Please refer to Weaknesses.

---

> ### Author Response · Authors · 2025-11-21
> **Response to Reviewer vr99**
>
> Thanks for your constructive comments. We are glad to answer all your questions.
>
> **W1: Why do you only introduce an uncertainty component?**
>
> > **R: In fact, a single uncertainty component  is a minimal, effective choice balancing privacy and learning.**  We explain the advantages of introducing one uncertainty in **lines 160–162 of Section 3.2** and in Figure 1. **In addition, we dedicate a separate subsection, ‘Superiority of Uncertain Similarity Data,’ within Section 3.2** to emphasize that incorporating a single uncertain component provides an optimal balance between privacy protection and learning performance, as stated in lines 177–180. Introducing one uncertainty component already prevents deterministic pairwise inference. **This is sufficient to prevent immediate deterministic inference of a paired label while retaining informative relational structure for learning** (related experiments results can be found in Tables 1-3 and appendix F).
>
>
>
> **W1: Are more uncertainty components better to mitigate the privacy risks?**
>
> > **R: Yes,** more uncertainty components increase uncertainty and thereby enhance privacy.
> >
> > **But, adding more components weakens the supervised information, making it more difficult to learn an effective classifier.** In addition, adding more uncertainty components also makes it more challenging to obtain an unbiased estimator.
>
>
>
> **W2: Some empirical results of privacy risks or the analysis about the probability of privacy risks**
>
> > **R: In fact, we have provided additional privacy protection analyses in Appendix I and Table 9.** In Table 9, we present the analysis comparing traditional similarity-based methods with the proposed USimUL definition and evaluating its effectiveness in privacy protection.
> >
> > **Besides, we add an experiment to further demonstrate the effectiveness of the proposed USimUL in privacy protection.** The detailed experimental settings and the complete set of results are provided in Appendix F.6 of the revised version. For ease of reading, we present only a subset of the results here.
> >
> > For the triplet dataset (A, B, C), we simulate five attack models as follows:
> >
> > (1) **A → B**: Given 30% label leakage of sample A, the attacker infers the label of the sample B based on the binding relationship.
> >
> > (2) **A → C**: Given 30% label leakage of sample A, the attacker infers the label of the sample C.
> >
> > (3) **B → A**: Given 30% label leakage of sample B, the attacker infers the label of the sample A.
> >
> > (4) **C → A**: Given 30% label leakage of sample C, the attacker infers the label of the sample A.
> >
> > (5) **B & C → A**: Given 30% label leakage of samples B & C, the attacker infers the label of the bound sample A.
> >
> > Here, we report baseline (similarity-based methods) leak rate, our method’s leak rate, and our method’s protection rate (over 5 trials). **The results on CIFAR-10 and SVHN show that label leak rate drops from 100%  to ≈50%**.  It is nearly equivalent to random guessing in the context of a binary classification task, which demonstrates that USimUL can effectively alleviate the privacy risk in similarity-based weakly supervised learning.
>
> CIFAR-10:
>
> |Init Leak Ratio|Scenario|#Init Leaked|#Similar at Risk (Baseline)|#Actually Leaked (Baseline)|#Actually Leaked (Ours)|Baseline Leak Rate (%)|Ours Leak Rate (%)|Ours Protected (%)|
> |-|-|-|-|-|-|-|-|-|
> |**30%**|(1)A→B|1800|1800|1800|884.88±18.36|100|49.16±1.02|50.84±1.02|
> ||(2)A→C|1800|1800|1800|898.74±16.38|100|49.93±0.91|50.07±0.91|
> ||(3)B→A|1800|1800|1800|902.52±7.20|100|50.14±0.40|49.86±0.40|
> ||(4)C→A|1800|1800|1800|902.34±19.44|100|50.13±1.08|49.87±1.08|
> ||(5)B&C→A|3600|3600|3600|888.12±21.78|100|24.67±0.61|75.33±0.61|
>
> SVHN:
>
> |Init Leak Ratio|Scenario|#Init Leaked|#Similar at Risk (Base)|#Actually Leaked (Base)|#Actually Leaked (Ours)|Baseline Leak Rate (%)|Ours Leak Rate (%)|Ours Protected (%)|
> |-|-|-|-|-|-|-|-|-|
> |**30%**|(1)A→B|2400|2400|2400|1203.12±21.12|100|50.13±0.88|49.87±0.88|
> ||(2)A→C|2400|2400|2400|1195.92±19.44|100|49.83±0.81|50.17±0.81|
> ||(3)B→A|2400|2400|2400|1214.16±15.36|100|50.59±0.64|49.41±0.64|
> ||(4)C→A|2400|2400|2400|1180.32±18.00|100|49.18±0.75|50.82±0.75|
> ||(5)B&C→A|4800|4800|4800|1205.52±26.64|100|25.12±0.56|74.88±0.56|

---

### Author Response · Authors · 2025-11-21
**The revised paper is uploaded**

**Thanks!**

We would like to express our sincere gratitude to all reviewers for their time and valuable comments! We are delighted that the reviewers recognize the value of our work and provided positive feedback regarding our clear motivation (`vr99`, `PaMN`, `E4fo`), theoretical contributions (`PaMN`, `2JGU`, `E4fo`), novel methodological design (`vr99`, `2JGU`), comprehensive experiments (`vr99`, `2JGU`, `E4fo`), and clear and well-structured writing (`vr99`, `2JGU`). We truly appreciate the reviewers' encouraging feedback and constructive suggestions, which have helped us refine our work.



**Modification in manuscript:**

Additional analyses and experiments are included, with revisions highlighted in blue. The modifications are as follows:

* **(Sec. 1 Line 42-43, Sec. 2.1 Line 107-109)**: Clarified the motivation. (Reviewer `2JGU`, C2)
* **(Sec. 3.3 Line 272-292, Table 4, Figure 4, Appendix F.7 Line 1198-1204, Algorithm 2 Line 1259-1267, Table 11 and Table 12):** Added estimation of class prior and experiments. (Reviewer `PaMN`, W2)
* **(Sec. 4.1 Line 405-409):** Provided more details of the unlabeled sample setting. (Reviewer `E4fo`, Q4)
* **(Sec. 4.1 Line 433-434):** Clarified the $\phi(z)$. (Reviewer `E4fo`, Q3)
* **(Table 4):** Added experimental results with larger inaccurate training class prior. (Reviewer `E4fo`, Q6)
* **(Appendix F.6 Line 1077-1195, Tables 9 and Table 10):** Added empirical results of privacy preserving performances of USimUL. (Reviewer `vr99`,W1 and `2JGU`, C1)

We have also provided detailed responses to all the reviewers' questions. We hope that these responses address the reviewer’s concerns and questions. Thank you again for your time and helpful comments. Have a good day!

---

### Meta-Review · Area_Chair_aXZs · 2026-01-06

**Summary:**

This manuscript propose USimUL, which replaces precise pair supervision with an uncertain triplet signal. They derive an unbiased risk estimator for this setting, give a generalization bound, and report experiments on several standard datasets plus a privacy-sensitive one. Reviewer scores are quite split, ranging from accept to borderline. So there’s clear disagreement: some reviewers like the framing and theory, others are worried about whether the setup actually works in practice and whether the privacy story is strong enough.

On the positive side, reviewers generally agree the motivation is valid, the estimator derivation is technically solid, and the experiments are reasonably broad and show decent performance. But the main reasons for rejection are following. First, the paper doesn’t really convince reviewers that the annotation protocol is feasible. It’s not clear how an annotator can say “at least two of these three are same-class” without basically finding a similar pair first, which could make the supervision more expensive than the thing it’s trying to replace. Second, the privacy improvement is mostly argued qualitatively. Multiple reviewers asked for a more concrete privacy evaluation or threat model; adding one unlabeled sample may not actually raise the bar much for an attacker. Third, the framework seems to rely on knowing class priors, and while the paper tests mild prior mismatch, it doesn’t explain how priors would be obtained in this new supervision regime, which makes the end-to-end story feel incomplete. There are also a few detail gaps in the writeup and experimental setup that hurt confidence.

After reading the paper and the reviews, I recommend reject. The idea is interesting and the math seems fine, but the paper doesn’t yet make a persuasive case that USimUL is practical to annotate and meaningfully privacy-preserving in a realistic pipeline. If the authors can clarify the annotation workflow, add a clearer privacy evaluation, and address the class-prior dependency in a principled way, this could be much stronger.

**Reviewer Concerns:**

The rebuttal clarified parts of the writing and did a better job explaining the intended privacy motivation and why the triplet style supervision is supposed to reduce direct label leakage. But the rebuttal still lacks a convincing, quantitative story on how effective the privacy protection actually is under a reasonable threat model. The assumptions also remain a bit unrealistic in practice, especially the reliance on information like class priors without a clear way to obtain them in the proposed setting.

**Reviewer Scores:**

Reviewers vr99 and PaMN did not response the rebuttal. So, I think that they will not change their scores.

Reviewer 2JGU gave 8, and I think that he/she will keep the score.

Reviewer E4fo stated that his/her primary focus remains on the effectiveness of USimUL in scenarios involving multiple unlabeled samples. so, i think that he/she will not raise the score.

---

### Decision · Program_Chairs · 2026-01-26

Reject